# TopWORDS-Poetry: Simultaneous Text Segmentation and Word Discovery for Classical Chinese Poetry via Bayesian Inference

**Changzai Pan**
Center for Statistical
Science & Department of
Industrial Engineering,
Tsinghua University
pcz18@mails.tsinghua.edu.cn

**Feiyue Li**
Department of Chinese
Language and Literature,
School of Humanities,
Tsinghua University
lifeiyue@tsinghua.edu.cn

**Ke Deng**[*]
Center for Statistical
Science & Department of
Industrial Engineering,
Tsinghua University
kdeng@tsinghua.edu.cn

## Abstract

As a precious cultural heritage of human beings, classical Chinese poetry has a very unique writing style and often contains special words that rarely appear in general Chinese texts, posting critical challenges for natural language processing. Little effort has been made in the literature for processing texts from classical Chinese poetry. This study fills in this gap with TopWORDS-Poetry, an unsupervised method that can achieve reliable text segmentation and word discovery for classical Chinese poetry simultaneously without pre-given vocabulary or training corpus. Experimental studies confirm that TopWORDS-Poetry can successfully recognize unique poetry words, such as named entities and literary allusions, from metrical poems of 《全唐诗》 (*Complete Tang Poetry*) and segment these poetry lines into sequences of meaningful words with high quality.

## 1 Introduction

Expressing complicated thoughts and feelings in refined language, poetry is the most shining pearl on the crown of human literature. As a special literary style, poetry has unique characteristics in rhythm, metre and language use, making it a critical challenge to process poetry texts via algorithms (Agirrezabal et al., 2016; Fang et al., 2009; Ahmed and Trausan-Matu, 2017).

Rising in the Spring and Autumn Period (770-476 B.C.) and flourishing in the Tang Dynasty (618-907 A.D.), classical Chinese poetry has a long history and profound cultural heritage. Composed of nearly 50,000 poems by over 2000 poets of the Tang Dynasty, 《全唐诗》 (*Complete Tang Poetry*, or CTP) displayed in Fig 1 (a) is the ideal object for studying classical Chinese poetry. Fig 1 (b) and (c) visualize the 200 poets with the most poems and the 300 most frequent Chinese characters in CTP.

Among the nearly 50,000 poems in CTP, 34,227 are metrical poems that have recognizable patterns

of syllables. Containing 1.3 million Chinese characters in total, the collection of metrical poems in CTP (referred to as $\mathcal{P}$ hereinafter) is the most essence of Tang poetry and our focus in this study. A typical metrical poem in CTP is composed of either 4 or 8 lines, each of which contains either 5 or 7 Chinese characters, and can be classified into 4 categories $\mathcal{P}_{4\times5}$, $\mathcal{P}_{4\times7}$, $\mathcal{P}_{8\times5}$ and $\mathcal{P}_{8\times7}$, where $\mathcal{P}_{m\times n}$ represents the poems with $m$ lines and $n$ Chinese characters in each line. In $\mathcal{P}$, there are 3,225 poems in $\mathcal{P}_{4\times5}$, 9,820 poems in $\mathcal{P}_{4\times7}$, 13,536 poems in $\mathcal{P}_{8\times5}$, and 7,646 poems in $\mathcal{P}_{8\times7}$. Fig 1 (d)-(g) show 4 famous poems (each for one of the 4 categories) by the famous poet Du Fu (杜甫, 712-770 A.D.) of the Tang Dynasty, with the word boundaries (highlighted by symbol '/') manually labelled by domain experts. For the majority of poems in CTP, however, no word boundaries are labelled yet.

The absence of word boundaries posts critical challenges in literary study of classical Chinese poetry at a large scale, and leads to a great appeal to develop efficient algorithms to achieve reliable text segmentation and word discovery for poetry texts(Li et al., 2016; Liu, 2019). However, such a task is very challenging, because classical Chinese poetry is composed of special domain-specific texts with many unique features.

First, classical Chinese poetry has a very unique writing style and often contains special words that rarely appear in general Chinese texts, such as unique named entities and literary allusions. Moreover, to express complicated thoughts and feelings with few words, poets were often forced to compress long phrases into shorter "pseudo words" for putting them into short poetry lines. Although these "pseudo words" are phrases instead of words from the grammatical point of view, many of them are treated as "poetry words" in practice because they have specific meanings and function as words (Liu, 1966; Li et al., 2016). Due to these facts, word discovery becomes a critical problem in processing

---

* Corresponding author.

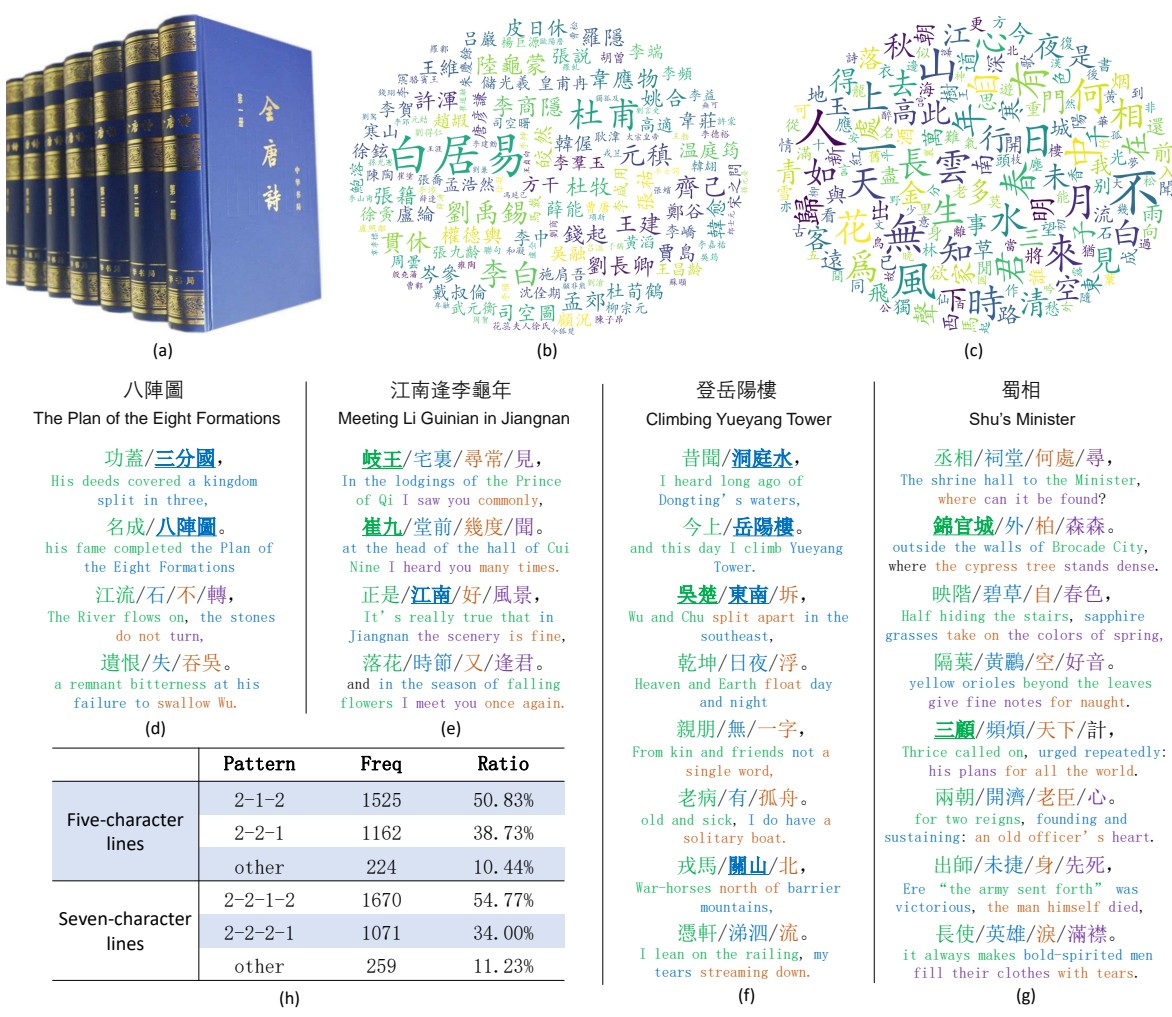

| | 八陣圖 | 江南逢李龜年 | 登岳陽樓 | 蜀相 |
| | The Plan of the Eight Formations | Meeting Li Guinian in Jiangnan | Climbing Yueyang Tower | Shu's Minister |

**八陣圖**
The Plan of the Eight Formations

功蓋/三分國，
His deeds covered a kingdom split in three,
名成/八陣圖。
his fame completed the Plan of the Eight Formations
江流/石/不/轉，
The River flows on, the stones do not turn,
遺恨/失/吞吳。
a remnant bitterness at his failure to swallow Wu.

(d)

**江南逢李龜年**
Meeting Li Guinian in Jiangnan

岐王/宅裏/尋常/見，
In the lodgings of the Prince of Qi I saw you commonly,
崔九/堂前/幾度/聞。
at the head of the hall of Cui Nine I heard you many times.
正是/江南/好/風景，
It's really true that in Jiangnan the scenery is fine,
落花/時節/又/逢君。
and in the season of falling flowers I meet you once again.

(e)

**登岳陽樓**
Climbing Yueyang Tower

昔聞/洞庭水，
I heard long ago of Dongting's waters,
今上/岳陽樓。
and this day I climb Yueyang Tower.
吳楚/東南/坼，
Wu and Chu split apart in the southeast,
乾坤/日夜/浮。
Heaven and Earth float day and night
親朋/無/一字，
From kin and friends not a single word,
老病/有/孤舟。
old and sick, I do have a solitary boat.
戎馬/關山/北，
War-horses north of barrier mountains,
憑軒/涕泗/流。
I lean on the railing, my tears streaming down.

(f)

**蜀相**
Shu's Minister

丞相/祠堂/何處/尋，
The shrine hall to the Minister, where can it be found?
錦官城/外/柏/森森。
outside the walls of Brocade City, where the cypress tree stands dense.
映階/碧草/自/春色，
Half hiding the stairs, sapphire grasses take on the colors of spring,
隔葉/黃鸝/空/好音。
yellow orioles beyond the leaves give fine notes for naught.
三顧/頻煩/天下/計，
Thrice called on, urged repeatedly: his plans for all the world
兩朝/開濟/老臣/心。
for two reigns, founding and sustaining: an old officer's heart.
出師/未捷/身/先死，
Ere "the army sent forth" was victorious, the man himself died,
長使/英雄/淚/滿襟。
it always makes bold-spirited men fill their clothes with tears.

(g)

| | Pattern | Freq | Ratio |
|---|---|---|---|
| Five-character lines | 2-1-2 | 1525 | 50.83% |
| | 2-2-1 | 1162 | 38.73% |
| | other | 224 | 10.44% |
| Seven-character lines | 2-2-1-2 | 1670 | 54.77% |
| | 2-2-2-1 | 1071 | 34.00% |
| | other | 259 | 11.23% |

(h)

Figure 1: Illustration of classical Chinese poetry in 《全唐诗》 (*Complete Tang Poetry*, or CTP). (a) The book series of CTP. (b) The 200 poets with the most poems in CTP. (c) The 300 most frequent characters in CTP. (d)-(g): Four famous poems by the famous poet Du Fu (杜甫, 712-770 A.D.) belonging to $\mathcal{P}_{4\times5}$, $\mathcal{P}_{4\times7}$, $\mathcal{P}_{8\times5}$ and $\mathcal{P}_{8\times7}$ respectively, with word boundaries (highlighted by symbol '/') manually labelled by domain experts. (h) Major segmentation patterns of metrical poems in CTP discovered based on 1,000 manually segmented poems that were randomly sampled from the pool of all metrical poems in CTP via stratification sampling.

classical Chinese poetry.

Second, to keep proper rhythm, text segmentation of classical Chinese poetry lines follows a few major segmentation patterns. Previous studies have revealed that 5-character poetry lines mainly follow two segmentation patterns 2-1-2 and 2-2-1, while 7-character poetry lines mainly follow another two segmentation patterns 2-2-1-2 and 2-2-2-1 (Chen, 1979; Yip, 1997; Feng, 2019; Liu, 2019). In this study, we refer to these 4 major segmentation patterns as $\varrho_{212}$, $\varrho_{221}$, $\varrho_{2212}$ and $\varrho_{2221}$, respectively. Apparently, $\varrho_{2212}$ and $\varrho_{212}$ correspond to the same segmentation pattern for poetry lines of different lengths, and form a general segmentation pattern $\varrho_{*12}$. Similarly, $\varrho_{2221}$ and $\varrho_{221}$ form another general segmentation pattern $\varrho_{*21}$. To investigate the

proportion of these segmentation patterns in $\mathcal{P}$, we randomly sampled 1,000 poems from $\mathcal{P}$ (referred to as $\mathcal{P}_{1000}$) by stratification sampling (i.e., 250 poems for each of the 4 poem types in $\mathcal{P}$), and manually segmented them as the demonstrating examples shown in Fig 1 (d)-(g). Figure 1 (h) summarizes the obtained results, confirming that most poetry lines indeed precisely follow these 4 segmentation patterns.

Third, on the other hand, however, about 10% of poetry lines violate these major segmentation patterns due to appearance of long words containing more than 2 Chinese characters, such as named entities and literary allusions. For example, in the 4 poems shown in Fig 1 (d)-(g), there are 5 poetry lines that violate the 4 major segmentation

patterns because of long words with 3 Chinese characters. Due to this phenomenon, text segmentation becomes a non-trivial issue.

These unique features of poetry texts make it inappropriate to directly apply popular tools for processing general Chinese texts based on supervised learning, such as Jieba (Sun, 2012), StanfordNLP (Manning et al., 2014), THULAC (Sun et al., 2016), PKUSEG (Luo et al., 2019), and LTP (Che et al., 2021), to process classical Chinese poetry. As we will show in the result Section, due to the lack of training for poetry texts, these methods suffer from serious performance degradation in text segmentation, and tend to miss many poetry words because of the limited ability in discovering unregistered words. And, it is not realistic to tune these tools for this specific task, because no qualified training corpus that contains enough well-segmented classical Chinese poetry lines is publicly available yet, to the best of our knowledge. Some methods based on unsupervised learning, e.g., TopWORDS (Deng et al., 2016), GTS (Yuan et al., 2020) and others (Goldwater et al., 2009; Mochihashi et al., 2009; Pearl et al., 2010), are strong in discovering unregistered words, but suffer from poor performance on text segmentation in the current scenario because they do not take the special features of poetry texts into consideration.

As pointed out by Pan et al. (2022), when processing domain-specific Chinese texts like poetry that potentially contain many unregistered words, word discovery and text segmentation become two critically important tasks entangled with each other. Most existing methods for Chinese text segmentation work under the support of a pre-loaded vocabulary, either explicitly given apriori or implicitly encoded in a pre-trained large language model. If the pre-loaded vocabulary covers the target corpus well, these methods typically perform reasonably well; once the underlying vocabulary of the target corpus exceeds the pre-given vocabulary seriously, however, we often observe significant performance degeneration of these methods, as demonstrated in Pan et al. (2022). Because classical Chinese poetry contains many unique named entities and literary allusions that never appear in general texts and follow specific segmentation templates, methods without considering these features tend to miss many poetry words and segment poetry lines in an inappropriate manner.

In the literature, few methods have been pro-

posed for processing classical Chinese poetry. Li et al. (2016) proposed to recognize meaningful words in classical Chinese poetry by filtering out fake words from word candidates via a modified support vector machine trained by manually labelled training samples. This method relies heavily on manually labelled samples to capture features of poetry words, and completely ignores the segmentation patterns of poetry lines. Liu (2019) proposed to formulate the text segmentation problem of a poetry line into a pattern selection problem under the assumption that every poetry line must precisely follow one of the two general segmentation patterns $\varrho_{*21}$ or $\varrho_{*12}$, and achieve text segmentation by selecting the plausible pattern based on only the last three characters of each poetry line via weighted point-wise mutual information. Although such a method takes good advantage of the segmentation patterns and performs reasonably well, it oversimplifies the problem via a restrictive assumption and tends to miss all named entities and literary allusions with more than 2 Chinese characters.

To better resolve this challenging problem, we need an algorithm that can wisely utilize the segmentation patterns of poetry lines, is flexible enough to deal with situations where the regular segmentation patterns are violated, and has strong word discovery ability to recognize special poetry words with little training information. In this study, we fill in this gap with TopWORDS-Poetry, an algorithm with all these desired features. Extending the Bayesian framework of TopWORDS-Seg proposed by Pan et al. (2022) to a more sophisticated hierarchical Bayesian model to reflect the constraints on text segmentation due to the major segmentation patterns, TopWORDS-Poetry is able to achieve effective text segmentation and word discovery simultaneously for classical Chinese poems with metrical patterns.

An experiment study on metrical poems in CTP confirms that TopWORDS-Poetry can correctly segment these poetry texts, with various named entities, literary allusions and special poetry words effectively recognized. Compared to existing methods in the literature, TopWORDS-Poetry is superior on processing classical Chinese poetry with a significant performance margin. The outputs of TopWORDS-Poetry may promote the applications of AI approaches in the study of classical Chinese literature and inspire more research efforts to study classical Chinese poetry via the paradigm of digital

humanity.

## 2 TopWORDS-Poetry

Proposed by Pan et al. (2022), TopWORDS-Seg is an effective method for processing open-domain Chinese texts that can achieve high-quality text segmentation and word discovery simultaneously. Modelling the target texts by the uni-gram language model with an over-complete initial dictionary $\mathcal{D}$ obtained by enumerating all qualified word candidates, TopWORDS-Seg utilizes the word boundaries suggested by a pre-given text segmenter (e.g., PKUSEG) as prior information to guide usage frequency estimation and pruning of word candidates in $\mathcal{D}$ under a Bayesian framework, and finally achieves text segmentation based on the pruned word dictionary with respect to the prior word boundaries.

As demonstrated by Pan et al. (2022), the Bayesian framework of TopWORDS-Seg integrates the advantages of a strong word discoverer TopWORDS and an effective text segmenter PKUSEG, leading to a stronger word discoverer and a more robust text segmenter for open-domain Chinese texts.

In this section, we extend the Bayesian framework of TopWORDS-Seg with a more flexible way to utilize prior information on text segmentation. Instead of using the word boundaries suggested by a pre-given text segmenter, e.g., PKUSEG, as prior information to guide learning, we encode the constraints on text segmentation due to various segmentation patterns (i.e., $\varrho_{*12}$ and $\varrho_{*21}$) into a more sophisticated prior distribution, coming up with a hierarchical Bayesian model to infer the unknown segmentation pattern of each poetry line.

### 2.1 The Bayesian Framework

Following the notations in Pan et al. (2022), we let $\mathcal{T} = \{T_1, \cdots, T_n\}$ be the unsegmented Chinese poetry lines of interest, $\mathcal{A} = \{a_1, \cdots, a_M\}$ be the set of Chinese characters that appears in $\mathcal{T}$, and $\mathcal{D}_{\mathcal{T}}$ be the underlying vocabulary of $\mathcal{T}$. For each poetry line $T_j$ of length $L_j$, let $B_j = (b_{j1}, \ldots, b_{jL_j})$ be its word boundary profile, where $b_{jl} = 1$ means there is a word boundary behind the $l$th character of $T_j$, and $b_{jl} = 0$ otherwise. Apparently, $B_j$ determines how $\mathcal{T}_j$ is segmented. Our goal is to learn the unknown vocabulary $\mathcal{D}_{\mathcal{T}}$ and word boundary profiles $\boldsymbol{B} = \{B_1, \cdots, B_n\}$ from $\mathcal{T}$.

To discover $\mathcal{D}_{\mathcal{T}}$, we first construct an over-

complete initial dictionary $\mathcal{D} = \{w_1, w_2, \ldots, w_N\}$ by enumerating all qualified word candidates in $\mathcal{T}$. This can be achieved in three steps: first, we enumerate all possible sub-strings in $\mathcal{T}$ whose length $\leq \tau_L$ and frequency $\geq \tau_F$, as suggested by Top-WORDS, to form a collection of word candidates denoted as $\mathcal{D}_E$; second, we segment each poetry line $\mathcal{T}_j$ according to the two general segmentation patterns $\varrho_{*21}$ and $\varrho_{*12}$ respectively to get a collection of $\mathcal{T}_j$-specific word candidates $\mathcal{D}_j$, and assemble them into an additional collection of word candidates $\mathcal{D}_P = \cup_{j=1}^n \mathcal{D}_j$; finally, we unite $\mathcal{D}_E$ and $\mathcal{D}_P$ to get $\mathcal{D} = \mathcal{D}_E \cup \mathcal{D}_P$. Throughout this paper, we assume that the initial dictionary $\mathcal{D}$ generated in this way covers the underlying vocabulary $\mathcal{D}_{\mathcal{T}}$ of interest, i.e., $\mathcal{D}_{\mathcal{T}} \subseteq \mathcal{D}$.

Define $\boldsymbol{\theta} = \{\theta_w\}_{w \in \mathcal{D}}$, where $\theta_w$ is the usage probability of word candidate $w$ in $\mathcal{T}$. Under the uni-gram language model with $\mathcal{D}$ as the vocabulary, we have the following likelihood function for poetry lines in $\mathcal{T}$ when $\mathcal{D}$, $\boldsymbol{B}$ and $\boldsymbol{\theta}$ are given:

$$
\begin{aligned}
\mathbb{P}(\mathcal{T} \mid \mathcal{D}; \boldsymbol{\theta}, \boldsymbol{B}) &= \prod_{j=1}^n \mathbb{P}(T_j \mid \mathcal{D}; \boldsymbol{\theta}, B_j) \\
&= \prod_{j=1}^n \prod_{w \in \mathcal{D}} \theta_w^{n_w(T_j, B_j)}, \quad (1)
\end{aligned}
$$

where $n_w(T_j, B_j)$ is the number of appearance of $w$ in $T_j$ under segmentation $B_j$.

To learn the unknown model parameters $(\boldsymbol{\theta}, \boldsymbol{B})$ via Bayesian inference, we follow Pan et al. (2022) to adopt the following independent prior:

$$
\pi(\boldsymbol{\theta}, \boldsymbol{B}) = \pi(\boldsymbol{\theta}) \cdot \pi(\boldsymbol{B}) = \pi(\boldsymbol{\theta}) \cdot \prod_{j=1}^n \pi(B_j),
$$

where

$$
\pi(\boldsymbol{\theta}) = Dirichlet(\boldsymbol{\theta} \mid \boldsymbol{\alpha}) \quad (2)
$$

is the same conjugate prior as in TopWORDS-Seg, with $\boldsymbol{\alpha} = \{\alpha_w\}_{w \in \mathcal{D}}$ being the parameter vector of a $n$-dimensional Dirichlet distribution. Here, we choose to specify $\alpha_w = 1$ for all $w \in \mathcal{D}$, leading to a non-informative plat prior distribution for $\boldsymbol{\theta}$, as suggested by Pan et al. (2022). The prior distribution of $\boldsymbol{B}$, i.e., $\pi(\boldsymbol{B}) = \prod_{j=1}^n \pi(B_j)$, however, needs to be properly specified to fit the special features of poetry lines.

Different from a general Chinese sentence, whose segmentation usually does not follow obvious patterns, a poetry line in this study typically follows a few major segmentation patterns as we

have pointed out previously. To be concrete, let $\mathscr{P}_L$ be the collections of segmentation patterns for poetry lines of length $L$, and $B_\varrho = (b_{\varrho 1}, \cdots, b_{\varrho L})$ be the word boundary profile of a segmentation pattern $\varrho \in \mathscr{P}_L$, where $b_{\varrho,l} = 1$ if $\varrho$ puts a word boundary behind the $l$-th Chinese character of a poetry line, and 0 otherwise. In this study, we have two collections of segmentation patterns

$$\mathscr{P}_5 = \{\varrho_{221}, \varrho_{212}\} \text{ and } \mathscr{P}_7 = \{\varrho_{2221}, \varrho_{2212}\},$$

with the following word boundary profiles:

$$B_{\varrho 212} = (0,1,1,0,1), \; B_{\varrho 2212} = (0,1,0,1,1,0,1),$$

$$B_{\varrho 221} = (0,1,0,1,1), \; B_{\varrho 2221} = (0,1,0,1,0,1,1).$$

To reflect the impact of these segmentation patterns on text segmentation, we use the following mixture distribution as the prior distribution of $B_j$:

$$\pi(B_j) = \sum_{\varrho \in \mathscr{P}_{L_j}} \lambda_\varrho \cdot \pi(B_j|\varrho), \qquad (3)$$

where $\boldsymbol{\lambda}_{\mathscr{P}_{L_j}} = \{\lambda_\varrho\}_{\varrho \in \mathscr{P}_{L_j}}$ is a probability vector defined over segmentation patterns in $\mathscr{P}_{L_j}$, and

$$\pi(B_j|\varrho) = \prod_{l=1}^{L_j} Binary(b_{jl}|\rho_{\varrho l})$$

with $\rho_{\varrho l}$ being a $\varrho$-specific probability value determined by $B_\varrho$, the word boundary profile of segmentation pattern $\varrho$. Here, we specify

$$\rho_{\varrho l} = \begin{cases} (1-\kappa) \cdot b_{\varrho l} + \kappa \cdot \rho, & l < L_\varrho, \\ 1, & l = L_\varrho, \end{cases} \qquad (4)$$

where $\rho \in (0,1)$ is the probability of placing a word boundary at each position randomly by a pseudo segmenter, and $\kappa \in (0,1)$ is a smoothing parameter to mix the segmentation pattern $\varrho$ and the pseudo segmenter. In this study, we set $\rho = 0.5$ by default, and leave $\kappa$ as a tunable parameter. Apparently, a $\kappa$ closer to 0 means a stronger prior preference to follow the segmentation pattern $\varrho$. Such a prior setting sufficiently considers the constraints in segmenting poetry lines due to the segmentation patterns, while gives us the flexibility to violate these constraints, leading to an ideal framework for processing classical Chinese poetry.

Because there are two collections of segmentation patterns, i.e., $\mathscr{P}_5$ and $\mathscr{P}_7$, in this study, we need two groups of parameters $\boldsymbol{\lambda}_{\mathscr{P}_5}$ and $\boldsymbol{\lambda}_{\mathscr{P}_7}$ to reflect the prior proportion of different segmentation

patterns in $\mathscr{P}_5$ and $\mathscr{P}_7$, respectively. In case that the proportion of segmentation patterns in $\mathscr{P}$ is precisely known, we can simply specify $\boldsymbol{\lambda}_{\mathscr{P}}$ to the known proportion vector to reflect the prior knowledge. In practice, however, we usually do not have such knowledge. In such a scenario, a reasonable strategy is to specify the following conjugate prior distribution for $\boldsymbol{\lambda}_{\mathscr{P}_5}$ and $\boldsymbol{\lambda}_{\mathscr{P}_7}$,

$$\pi(\boldsymbol{\lambda}_{\mathscr{P}_5}) = Dirichlet(\boldsymbol{\lambda}_{\mathscr{P}_5} \mid \boldsymbol{\beta}_{\mathscr{P}_5}), \qquad (5)$$

$$\pi(\boldsymbol{\lambda}_{\mathscr{P}_7}) = Dirichlet(\boldsymbol{\lambda}_{\mathscr{P}_7} \mid \boldsymbol{\beta}_{\mathscr{P}_7}), \qquad (6)$$

and infer them along the learning process. Such a Bayesian framework leads to a hierarchical Bayesian model (Good et al., 1966; Gelman et al., 2013) for $B_j$'s with $(\boldsymbol{\beta}_{\mathscr{P}_5}, \boldsymbol{\beta}_{\mathscr{P}_7}, \kappa)$ as hyperparameters. Here, we choose to specify $\boldsymbol{\beta}_{\mathscr{P}_5}$ and $\boldsymbol{\beta}_{\mathscr{P}_7}$ with constant vectors whose elements all equal to 1.

Finally, we come up with the following joint prior distribution for $\boldsymbol{\theta} = \{\theta_w\}_{w \in \mathcal{D}}$, $\boldsymbol{\lambda} = (\boldsymbol{\lambda}_{\mathscr{P}_5}, \boldsymbol{\lambda}_{\mathscr{P}_7})$ and $\boldsymbol{B} = \{B_j\}_{1 \leq j \leq n}$:

$$\pi(\boldsymbol{\theta}, \boldsymbol{\lambda}, \boldsymbol{B}) = \pi(\boldsymbol{\theta}) \cdot \pi(\boldsymbol{\lambda}) \cdot \prod_{j=1}^{n} \pi(B_j \mid \boldsymbol{\lambda}_{\mathscr{P}_{L_j}}),$$

where

$$\pi(\boldsymbol{\lambda}) = \pi(\boldsymbol{\lambda}_{\mathscr{P}_5}) \cdot \pi(\boldsymbol{\lambda}_{\mathscr{P}_7}),$$

with $\pi(\boldsymbol{\theta})$, $\pi(B_j|\boldsymbol{\lambda}_{\mathscr{P}_{L_j}})$ and $\pi(\boldsymbol{\lambda}_{\mathscr{P}})$ defined in Eq. (2)-(6), respectively. Accordingly, we have the following posterior distribution of $(\boldsymbol{\theta}, \boldsymbol{\lambda}, \boldsymbol{B})$ given $\mathcal{T}$ and $\mathcal{D}$ based on the Bayes theorem:

$$\begin{aligned} & f(\boldsymbol{\theta}, \boldsymbol{\lambda}, \boldsymbol{B} \mid \mathcal{D}, \mathcal{T}) \\ \propto \; & \pi(\boldsymbol{\theta}, \boldsymbol{\lambda}, \boldsymbol{B}) \cdot \mathbb{P}(\mathcal{T} \mid \mathcal{D}; \boldsymbol{\theta}, \boldsymbol{B}). \qquad (7) \end{aligned}$$

## 2.2 Word Discovery and Text Segmentation

Based on the posterior distribution in Eq. (7), word discovery and text segmentation can be achieved by following the strategy recommended by Deng et al. (2016).

First, we maximize the marginal posterior distribution below with $\boldsymbol{B}$ integrated out

$$f(\boldsymbol{\theta}, \boldsymbol{\lambda} \mid \mathcal{D}, \mathcal{T}) = \int f(\boldsymbol{\theta}, \boldsymbol{\lambda}, \boldsymbol{B} \mid \mathcal{D}, \mathcal{T}) d\boldsymbol{B}$$

by the EM algorithm (Dempster et al., 1977) to get the Maximum a Posteriori (MAP) estimate of $(\boldsymbol{\theta}, \boldsymbol{\lambda})$ below:

$$(\hat{\boldsymbol{\theta}}, \hat{\boldsymbol{\lambda}}) = \arg\max_{(\boldsymbol{\theta}, \boldsymbol{\lambda})} f(\boldsymbol{\theta}, \boldsymbol{\lambda} \mid \mathcal{D}, \mathcal{T}). \qquad (8)$$

Second, we get the significance score $\psi_w$ of each word candidate $w \in \mathcal{D}$ by calculating the likelihood ratio between the full model with all word candidates in $\mathcal{D}$ and the reduced model with $w$ removed from $\mathcal{D}$:

$$\psi_w = \log\left(\frac{\prod_{j=1}^n \mathbb{P}(T_j \mid \mathcal{D}; \hat{\boldsymbol{\theta}}, \hat{\boldsymbol{\lambda}})}{\prod_{j=1}^n \mathbb{P}(T_j \mid \mathcal{D}; \hat{\boldsymbol{\theta}}_{[w=0]}, \hat{\boldsymbol{\lambda}})}\right), \quad (9)$$

where $\hat{\boldsymbol{\theta}}_{[w=0]}$ is a modification of $\hat{\boldsymbol{\theta}}$ by setting $\hat{\theta}_w = 0$ with all the other elements of $\hat{\boldsymbol{\theta}}$ unchanged, and

$$\begin{aligned} &\mathbb{P}(T_j \mid \mathcal{D}; \hat{\boldsymbol{\theta}}, \hat{\boldsymbol{\lambda}}) \\ &= \sum_{B_j \in \mathcal{B}_j} \pi(B_j \mid \hat{\boldsymbol{\lambda}}) \cdot \mathbb{P}(T_j \mid \mathcal{D}; \hat{\boldsymbol{\theta}}, B_j), \\ &= \sum_{B_j \in \mathcal{B}_j} \sum_{\varrho \in \mathscr{P}_{L_j}} \hat{\lambda}_\varrho \cdot \pi(B_j \mid \varrho) \cdot \mathbb{P}(T_j \mid \mathcal{D}; \hat{\boldsymbol{\theta}}, B_j). \end{aligned}$$

Because a larger $\psi_w$ means that word candidate $w$ is more important for fitting the poetry texts and thus more likely to be a meaningful word, we achieve word discovery by keeping only word candidates whose $\psi_w \geq \tau_\psi$. Since $-2\psi_w \sim \chi^2$ asymptotically under the null hypothesis that the reduced model with $w$ removed is the true model, we set the threshold $\tau_\psi$ to the $(1 - \frac{0.05}{N})$-quantile of $\chi^2$ distribution by default, based on the Bonferroni correction principle for multiple hypothesis testing.

Third, based on the MAP estimate $(\hat{\boldsymbol{\theta}}, \hat{\boldsymbol{\lambda}})$ of the pruned word dictionary $\mathcal{D}$ obtained in the previous step, we can segment $T_j$ according to

$$\hat{B}_j = \max_{B \in \mathcal{B}_j} \mathbb{P}(B \mid \mathcal{D}, T_j; \hat{\boldsymbol{\theta}}, \hat{\boldsymbol{\lambda}}), \quad (10)$$

where $\mathcal{B}_j$ stands for the set of all possible word boundary profiles of $T_j$. A more robust strategy recommended by Deng et al. (2016) is to calculate

$$\gamma_{jl} = \sum_{B \in \mathcal{B}_j} b_{jl} \cdot \mathbb{P}(B_j \mid \mathcal{D}, T_j; \hat{\boldsymbol{\theta}}, \hat{\boldsymbol{\lambda}}), \quad (11)$$

and put a word boundary behind the $l$th position of $T_j$ if $\gamma_{jl}$ is larger than a pre-given threshold $\tau_S$.

## 2.3 Computational Issues

Although TopWORDS-Poetry follows a similar strategy as in TopWORDS-Seg to achieve word discovery and text segmentation, it involves more computational challenges due to the more complicated prior distribution $\pi(B_j)$, which is a mixture distribution with unknown parameter $\boldsymbol{\lambda}_{\mathscr{P}_{L_j}}$.

To ease the computation challenges, we propose to introduce an additional group of auxiliary latent variables $\boldsymbol{Z} = \{Z_j\}_{1 \leq j \leq n}$ with $Z_j$ standing for the underlying segmentation pattern of poetry line $T_j$. With the help of $\boldsymbol{Z}$, we can expand the target posterior distribution $f(\boldsymbol{\theta}, \boldsymbol{\lambda}, \boldsymbol{B} \mid \mathcal{D}, \mathcal{T})$ in Eq. (A.1) to a higher dimensional distribution $f(\boldsymbol{\theta}, \boldsymbol{\lambda}, \boldsymbol{Z}, \boldsymbol{B} \mid \mathcal{D}, \mathcal{T})$, which takes $f(\boldsymbol{\theta}, \boldsymbol{\lambda}, \boldsymbol{B} \mid \mathcal{D}, \mathcal{T})$ as its marginal distribution and is computationally more friendly. and thus implement computations based on the expanded posterior distribution instead of the original one. We leave all these computation details to Appendix A-C, with intuitions behind some key formulas explained in Appendix D.

## 2.4 TopWORDS-Poetry Pipeline

Assembling the above ingredients, we come up with the TopWORDS-Poetry algorithm composed of four stages: a dictionary initialization stage to generate the initial word dictionary $\mathcal{D}$, a prior specification stage to specify prior distributions based on pre-given segmentation patterns and hyperparameters, a word discovery stage to estimate $(\boldsymbol{\theta}, \boldsymbol{\lambda})$ and prune the initial word dictionary $\mathcal{D}$ into a more concise final dictionary $\mathcal{D}_F$, and a text segmentation stage to segment poetry lines based on the refined model. Figure 2 illustrates the algorithm pipeline of TopWORDS-Poetry.

In this pipeline, $\tau_L$, $\tau_F$ and $\kappa$ are three control parameters to be specified by the users. Since words with more than 3 Chinese characters are very rare in classical Chinese poetry and it is not realistic to discover rare words that appear only once, we set $\tau_L = 3$ and $\tau_F = 2$ by default. Following the strategy proposed by Pan et al. (2022), we choose a larger $\kappa$ in the word discovery stage (referred to as $\kappa_d$) to encourage the discovery of unregistered words, and a smaller $\kappa$ in the text segmentation stage (referred to as $\kappa_s$) to show more respect to the segmentation patterns. In this study, we find that TopWORDS-Poetry works well by setting $\kappa_d = 0.5$ and $\kappa_s = 0.001$, and thus use these values as the default setting.

Note that TopWORDS-Peotry is an unsupervised method in nature, because it only takes a set of segmentation patterns (i.e., $\mathscr{P}_5$ and $\mathscr{P}_7$) and a few hyper-parameters as inputs, and does not need any training corpus or pre-given vocabulary.

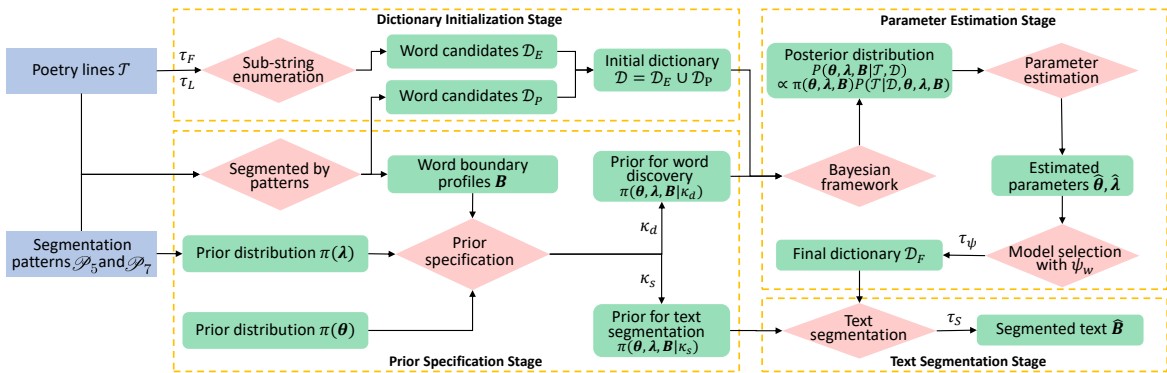

Figure 2: Pipeline of the TopWORDS-Poetry algorithm

## 2.5 Considering Correlation within a Couplet

In the TopWORDS-Poetry algorithm, we assume that the choice of segmentation pattern is completely independent across different poetry lines. In practice, however, a metrical poem with 4 or 8 lines is composed of 2 or 4 couplets, each of which contains two poetry lines with positively correlated segmentation patterns (see Fig E1 for detailed evidence). Such a fact means that we can slightly modify TopWORDS-Poetry for a more efficient algorithm by modelling the correlation accordingly. Hereinafter, we refer to the modified TopWORDS-Poetry algorithm as TopWORDS-Poetry*.

## 3 Experimental Study

We applied TopWORDS-Poetry on the poetry texts in $\mathcal{P}$, the collection of all metrical poems in CTP, under the default setting of $\tau_L = 3$, $\tau_F = 2$, $\rho = 0.5$, $\kappa_d = 0.5$, $\kappa_s = 0.001$, $\tau_\psi = 0.05$ and $\tau_S = 0.5$. For comparison purpose, we also applied 6 competing methods to the same data set, including Jieba (Sun, 2012), StanfordNLP (Manning et al., 2014), THULAC (Sun et al., 2016), PKUSEG (Luo et al., 2019), LTP (Che et al., 2021), TopWORDS (Deng et al., 2016) and TopWORDS-Seg under their default settings.

Unfortunately, however, no executable program is available for either the modified SVM method (referred to as ModSVM) for discovering poetry words by Li et al. (2016) or the pattern selection method (referred to as SelectPattern) for segmenting poetry texts by Liu (2019). Although it is straightforward to implement SelectPattern by ourselves, it is non-trivial to implement ModSVM due to many subtle details. Therefore, we only include SelectPattern in the experimental study. Moreover, Liu (2019) also proposed a baseline method (referred to as RandomPattern here) that randomly picks up a segmentation pattern from $\{\varrho_{*12}, \varrho_{*21}\}$ for each poetry line. We also include RandomPattern in our experiment.

Furthermore, we also evaluate the performance of GPT-style methods on segmenting classical Chinese poetry by asking ChatGPT (model version: gpt-35-turbo) to segment all poetry lines in the testing dataset $\mathcal{P}_{1000}$. Appendix F reports the prompt designing for calling ChatGPT for this task in detail.

## 3.1 Performance Evaluation Criteria

Because no gold standard is publicly available for text segmentation of poems in CTP, we utilize the manually segmented poems in $\mathcal{P}_{1000}$ as the test data for performance evaluation. Randomly sampled from $\mathcal{P}$, the pool of all metrical poems in CTP, via stratification sampling, $\mathcal{P}_{1000}$ covers 250 poems for each of the 4 poem types of interest and is a group of representative samples from $\mathcal{P}$. Let $\mathcal{T}_{test}$ be the indices of all poetry lines in $\mathcal{P}_{1000}$, $\boldsymbol{B}_{test} = \{B_j\}_{j \in \mathcal{T}_{test}}$ be the true word boundary profiles of poetry lines in $\mathcal{T}_{test}$, and $V_{test}$ be the true vocabulary of $\mathcal{T}_{test}$ obtained by collecting all words in the segmented poetry lines in $\mathcal{T}_{test}$.

For each competing method, let $V_d$ be the discovered vocabulary based on all poetry lines in $\mathcal{T}$, $\hat{\boldsymbol{B}} = \{\hat{B}_j\}_{j \in \mathcal{T}}$ be the predicted word boundary profiles, and $V_s$ be the collection of words that appear in the segmented poetry lines in $\mathcal{T}_{test}$ based on $\hat{\boldsymbol{B}}_{test} = \{\hat{B}_j\}_{j \in \mathcal{T}_{test}}$. We measure its performance on word discovery by *discovery recall* $R_d^{(t,l)}$ and *segmentation recall* $R_s^{(t,l)}$ defined as following:

$$R_d^{(t,l)} = \frac{|V_{test}^{(t,l)} \cap V_d^{(t,l)}|}{|V_{test}^{(t,l)}|}, \quad R_s^{(t,l)} = \frac{|V_{test}^{(t,l)} \cap V_s^{(t,l)}|}{|V_{test}^{(t,l)}|},$$

https://chat.openai.com/

| Method | Word discovery $(R_d^{(t,l)}, R_s^{(t,l)})$ | | | | | | Text segmentation $(F_S)$ | | | | |
|---|---|---|---|---|---|---|---|---|---|---|---|
| | $\|V_d\|$ | $R_d^{(2,2)}$ | $R_d^{(2,3)}$ | $\|V_s\|$ | $R_s^{(2,2)}$ | $R_s^{(2,3)}$ | $\mathcal{P}_{1000}$ | $\mathcal{P}_{250}^{4\times5}$ | $\mathcal{P}_{250}^{8\times5}$ | $\mathcal{P}_{250}^{4\times7}$ | $\mathcal{P}_{250}^{8\times5}$ |
| LTP | 39K | 59.33% | 57.32% | 2.5K | 19.28% | 29.94% | 20.00% | 21.16% | 18.72% | 21.72% | 19.65% |
| StanfordNLP | 77K | 77.04% | 63.69% | 6.0K | 43.56% | 45.54% | 39.49% | 40.45% | 40.26% | 41.07% | 37.80% |
| THULAC | 76K | 77.35% | 43.63% | 6.7K | 53.28% | 35.03% | 49.08% | 50.27% | 50.14% | 49.25% | 47.78% |
| PKUSEG | 83K | 81.82% | 48.09% | 7.6K | 59.26% | 36.31% | 52.17% | 52.62% | 52.96% | 52.65% | 51.20% |
| Jieba | 71K | 78.21% | 43.31% | 6.8K | 57.30% | 37.58% | 53.62% | 56.40% | 52.60% | 54.23% | 53.04% |
| TopWORDS | 54K | 77.96% | **89.81%** | 8.5K | 71.22% | **89.49%** | 63.71% | 63.34% | 62.49% | 65.08% | 64.07% |
| TopWORDS-Seg | 86K | 83.27% | 53.50% | 8.6K | 68.06% | 44.90% | 60.67% | 60.26% | 61.27% | 60.35% | 60.55% |
| RandomPattern | 96K | 95.12% | 0.00% | 9.7K | 81.61% | 0.00% | 69.09% | 63.62% | 64.70% | 72.80% | 72.56% |
| SelectPattern | 73K | 94.19% | 0.00% | 9.2K | 88.71% | 0.00% | 84.12% | 81.27% | 81.57% | 86.46% | 85.91% |
| ChatGPT | - | - | - | 8.0K | 77.12% | 66.24% | 73.48% | 72.19% | 73.17% | 71.99% | 74.96% |
| TopWORDS-Poetry | 82K | 96.19% | 83.76% | 9.3K | 88.98% | 62.74% | 84.48% | **81.50%** | 81.95% | 86.49% | 86.46% |
| TopWORDS-Poetry* | 82K | **96.25%** | 82.80% | 9.4K | **89.36%** | 56.69% | **84.71%** | 80.68% | **82.70%** | **86.85%** | **86.63%** |

Table 1: Performance of TopWORDS-Poetry on word discovery and text segmentation for classical Chinese poetry with comparison to competing methods.

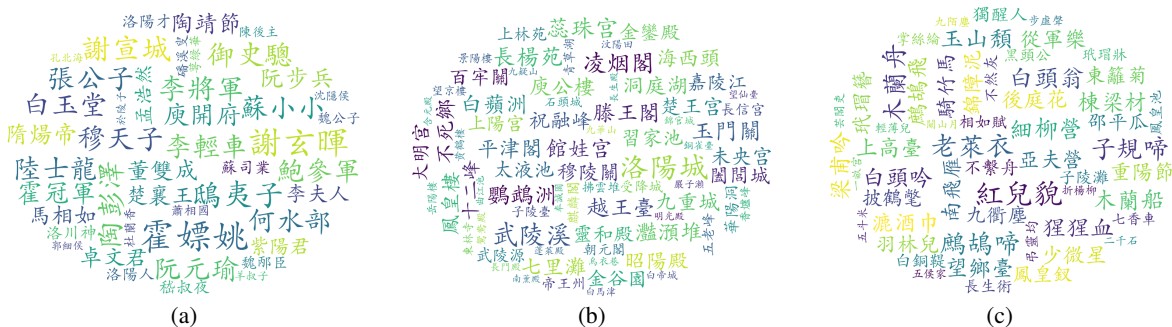

| (a) | (b) | (c) |
|---|---|---|

Figure 3: Results of the Top 100 most significant discovered names (a), addresses (b), and literary allusions (c) with more than 2 Chinese characters discovered by TopWORDS-Poetry.

where $t$ and $l$ are natural numbers, and

$$V^{(t,l)} = \{w \in V : freq(w) \geq t, len(w) \geq l\},$$

is a subset of $V$ composed of selected words, with $freq(w)$ being the raw frequency of string $w$ in $\mathcal{T}$ and $len(w)$ being the length of $w$. $R_d$ and $R_s$ evaluate word discovery from different perspectives, and provide us a more comprehensive view on this task in this application.

For test data $\mathcal{T}_{test}$, let $F_s$ be the $F_1$ score of $\hat{B}_{test}$ with respect to $B_{test}$. We measure the performance on text segmentation for $\mathcal{T}_{test}$ by $F_s$. In practice, by using poetry lines in $\mathcal{P}_{1000}$ and its subsets (e.g., $\mathcal{P}_{250}^{4\times5}$, $\mathcal{P}_{250}^{8\times5}$, $\mathcal{P}_{250}^{4\times7}$ and $\mathcal{P}_{250}^{8\times7}$, which cover different types of poems in it) as the test data, we can evaluate text segmentation performance at the overall as well as more detailed levels.

### 3.2 Results

It took 80 minutes in a cluster with 10 CPU cores (Platinum82, 2.5GHz) with a RAM of 28GB to finish all computations involved in this experiment. Table 1 compares the performance of proposed

| Patterns in $\mathcal{P}_5$ | Proportion | Patterns in $\mathcal{P}_7$ | Proportion |
|---|---|---|---|
| 2-1-2 | 55.88% | 2-2-1-2 | 59.57% |
| 2-2-1 | 35.71% | 2-2-2-1 | 30.92% |
| 2-3 | 7.37% | 2-2-3 | 8.82% |
| 3-2 | 1.03% | 2-3-2 | 0.69% |
| other | 0.01% | other | 0.00% |

Table 2: The distribution of segmentation patterns reported by TopWORDS-Poetry for poetry lines in $\mathcal{T}$.

methods on word discovery and text segmentation for the CTP data with 10 competing methods, from which we can see: (1) tools for processing general Chinese texts, such as LTP, Jieba and so on, perform poorly in this application; (2) TopWORDS recognizes most long words but at the price of relative low performance on short words, and cannot segment poetry lines well, with TopWORDS-Seg performing worse in most dimensions because the prior information provided by PKUSEG is misleading; (3) SelectPattern achieves good performance on segmenting poetry lines in general, but misses all long words with more than 2 Chinese characters and all irregular patterns for text segmentation

beyond the major patterns; (4) The $R_S$ and $F_S$ scores of ChatGPT concentrate in a narrow region of around 70%, leaving a wide performance margin (>10%) with respect to TopWORDS-Poetry. (5) TopWORDS-Poetry and TopWORDS-Poetry*, however, both achieve balanced performance on word discovery and text segmentation that outperforms all competing methods, with TopWORDS-Poetry* being slightly superior in most dimensions.

These results confirm that TopWORDS-Poetry is an effective word discoverer for poetry texts. Although $R_s^{(2,3)}$ of TopWORDS-Poetry, which is 62.74%, is much smaller than 89.49% reported by TopWORDS, it is the third largest value in the column and outperforms SelectPattern with a huge margin. We note that $R_s^{(2,3)} = 62.74\%$ means that TopWORDS-Poetry may miss 1/3 long words in $\mathcal{T}_{test}$ in the segmented texts. But, considering that $R_d^{(2,3)} = 83.76\%$ for TopWORDS-Poetry, we still capture the majority of these long words in the discovered word dictionary.

The results of TopWORDS-Poetry reported in Table 1 are under the default setting where hyper-parameter $\kappa_s$ is set to $\kappa_s = 0.001$. In this setting, TopWORDS-Poetry reports a relatively low $R_s^{(2,3)}$ score ($\sim$60%), with respect to the high $R_s^{(2,2)}$ score ($\sim$89%), because the low parameter $\kappa_s = 0.001$ discourages long words in text segmentation. In practice, however, we can adjust the granularity of poetry line segmentation in TopWORDS-Poetry by specifying hyper-parameter $\kappa_s$ differently: a larger $\kappa_s$ tends to reserve more long words in text segmentation, and thus leads to a higher chance to break through the restrictive segmentation templates. Therefore, if we could accept a little bit of decrease in the $F_S$ scores from the current $\sim$85% to $\sim$80%, we can get a $R_s^{(2,3)}$ that is as large as 80% by specifying $\kappa_s = 0.1$ instead.

Table 2 reports the segmentation patterns recognized by TopWORDS-Poetry from $\mathcal{T}$, confirming that TopWORDS-Poetry can successfully recognize rich segmentation patterns and correctly estimate their proportion. Figure 3 (a)-(c) visualize the most significant names, addresses and literary allusions with more than 2 Chinese characters discovered by TopWORDS-Poetry, suggesting that TopWORDS-Poetry indeed captures a lot of meaningful words that are often missed by most competing methods. These results confirm the superiority of the proposed method over competing methods and its ability as a powerful tool for processing classical Chinese poetry.

Additionally, we also report in Appendix G more detailed performance evaluation results on word discovery by word types (e.g., name, address, and literary allusion), where labelled words in test dataset $\mathcal{P}_{1000}$ are manually classified into different categories. Moreover, to investigate whether the advantage of TopWORDS-Poetry over the other methods is statistically significant, we further calculated the performance margin of TopWORDS-Poetry* between the second best competing methods in every performance evaluation dimension and tested the statistical significance of the obtained performance margin. Detailed results are reported in Appendix H.

## 4  Conclusions and Discussions

In this paper, we proposed TopWORDS-Poetry, a powerful unsupervised method for processing classical Chinese poetry that can segment poetry lines wisely with respect to a few major segmentation patterns, and recognize unique poetry words such as name entities and literary allusions successfully with little training information. Relying on a hierarchical Bayesian model to utilize prior information about known segmentation patterns of poetry lines, and rigorous statistical principles to achieve parameter estimation and model selection, TopWORDS-Poetry leads to an elegant solution to challenging NLP problems in an interpretable way.

We hope this proposed method can serve as a useful tool for scholars in the field of Chinese literary or digital humanity to conduct in-depth study on classical Chinese poetry via distant reading. For example, the literary allusions discovered from CTP by TopWORDS-Poetry may lead to very exciting research in near future, which may extend the research efforts discussed by Moritz et al. (2016); Sturgeon (2018); Burns et al. (2021). We also hope this work can promote more and more interactions between literary scholars and NLPers, and applications of NLP technology in cultural education and dissemination, helping more people to better enjoy the charm of Chinese traditional culture.

## Limitations

The proposed method relies on regular segmentation patterns in poetry lines to guide text segmentation and thus may fail to process poetry lines without such regular segmentation patterns. Moreover, the proposed method tends to miss rare words

that appear only one or two times in the target corpus due to its nature as an unsupervised approach, and may result in improper segmentation for poetry lines containing rare named entities.

## Acknowledgements

This research is partially supported by the National Scientific and Technological Innovation 2030 Major Project (No: 2020AAA0106501), the Guo Qiang Institute of Tsinghua University, and the Beijing Natural Science Foundation (Z190021). Changzai Pan was supported by China Scholarship Council during the early stage of this work. The authors would like to thank Miss Yafei Han, Miss Jiafei Song, Miss Linyue Zhang, Mr Yanggang Zhang, Mr Ning Cheng, Miss Xiting Song and other members in Ke Deng's lab and Feiyue Li's lab for their great help on generating manually labelled data for performance evaluation.

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

## A  EM Algorithm for Estimating $(\boldsymbol{\theta}, \boldsymbol{\lambda})$

By adding latent variables $\boldsymbol{Z}$, the target posterior distribution $f(\boldsymbol{\theta}, \boldsymbol{\lambda}, \boldsymbol{B} \mid \mathcal{D}, \mathcal{T})$ can be extended to $f(\boldsymbol{\theta}, \boldsymbol{\lambda}, \boldsymbol{Z}, \boldsymbol{B} \mid \mathcal{D}, \mathcal{T})$ as following:

$$
\begin{aligned}
&f(\boldsymbol{\theta}, \boldsymbol{\lambda}, \boldsymbol{Z}, \boldsymbol{B} \mid \mathcal{D}, \mathcal{T}) \\
=\ & \pi(\boldsymbol{\theta}) \cdot \pi(\boldsymbol{\lambda}) \cdot \prod_{j=1}^{n} \Big( \boldsymbol{\lambda}_{Z_j} \cdot \pi(B_j \mid Z_j) \\
& \cdot \mathbb{P}(\mathcal{T} \mid \mathcal{D}; \boldsymbol{\theta}, \boldsymbol{B}) \Big).
\end{aligned}
\tag{A.1}
$$

Given $(\boldsymbol{\theta}^{(t)}, \boldsymbol{\lambda}^{(t)})$, the current estimation of $(\boldsymbol{\theta}, \boldsymbol{\lambda})$, the E-step computes the Q-function below:

$$
\begin{aligned}
&Q(\boldsymbol{\theta}, \boldsymbol{\lambda}; \boldsymbol{\theta}^{(t)}, \boldsymbol{\lambda}^{(t)}) \\
=\ & \mathbb{E}\left( \log\left( \mathbb{P}(\boldsymbol{\theta}, \boldsymbol{\lambda}, \boldsymbol{B}, \boldsymbol{Z} \mid \mathcal{T}, \mathcal{D}) \right) \middle| \mathcal{D}, \mathcal{T}; \boldsymbol{\theta}^{(t)}, \boldsymbol{\lambda}^{(t)} \right) \\
=\ & C + \sum_{w \in \mathcal{D}} \left( \log \theta_w \cdot n_w(\boldsymbol{\theta}^{(t)}, \boldsymbol{\lambda}^{(t)}) \right) \\
& + \sum_{\varrho \in \mathscr{P}_5 \cup \mathscr{P}_7} \left( \log \lambda_\varrho \cdot n_\varrho(\boldsymbol{\theta}^{(t)}, \boldsymbol{\lambda}^{(t)}) \right),
\end{aligned}
$$

where $C$ is a constant that does not change with $\boldsymbol{\theta}$,

$$
n_w(\boldsymbol{\theta}^{(t)}, \boldsymbol{\lambda}^{(t)}) = \sum_{j=1}^{n} n_{w,j}(\boldsymbol{\theta}^{(t)}, \boldsymbol{\lambda}^{(t)}),
$$

$$
n_\varrho(\boldsymbol{\theta}^{(t)}, \boldsymbol{\lambda}^{(t)}) = \sum_{j=1}^{n} n_{\varrho,j}(\boldsymbol{\theta}^{(t)}, \boldsymbol{\lambda}^{(t)}),
$$

$$
\begin{aligned}
& n_{w,j}(\boldsymbol{\theta}^{(t)}, \boldsymbol{\lambda}^{(t)}) \\
=\ & \mathbb{E}(n_w(T_j, B_j) \mid \mathcal{D}, T_j; \boldsymbol{\theta}^{(t)}, \boldsymbol{\lambda}^{(t)}) \tag{A.2} \\
=\ & \sum_{B_j \in \mathcal{B}_j} n_w(T_j, B_j) \mathbb{P}(B_j \mid \mathcal{D}, T_j; \boldsymbol{\theta}^{(t)}, \boldsymbol{\lambda}^{(t)}),
\end{aligned}
$$

$$
\begin{aligned}
& n_{\varrho,j}(\boldsymbol{\theta}^{(t)}, \boldsymbol{\lambda}^{(t)}) \\
=\ & \mathbb{P}(Z_j = \varrho \mid \mathcal{D}, T_j; \boldsymbol{\theta}^{(t)}, \boldsymbol{\lambda}^{(t)}) \tag{A.3} \\
\propto\ & \lambda_\varrho^{(t)} \sum_{B_j \in \mathcal{B}_j} \pi(B_j \mid \varrho) \cdot \mathbb{P}(T_j \mid \mathcal{D}; \boldsymbol{\theta}^{(t)}, B_j).
\end{aligned}
$$

The M-step updates $(\boldsymbol{\theta}^{(t)}, \boldsymbol{\lambda}^{(t)})$ by maximizing $Q(\boldsymbol{\theta}, \boldsymbol{\lambda}; \boldsymbol{\theta}^{(t)}, \boldsymbol{\lambda}^{(t)})$ with respect to $(\boldsymbol{\theta}, \boldsymbol{\lambda})$, leading to the updating function below:

$$
\theta_w^{(t+1)} = \frac{n_w(\boldsymbol{\theta}^{(t)}, \boldsymbol{\lambda}^{(t)})}{\sum_{w \in \mathcal{D}} n_w(\boldsymbol{\theta}^{(t)}, \boldsymbol{\lambda}^{(t)})}, \ \forall\, w \in \mathcal{D}; \tag{A.4}
$$

$$
\lambda_\varrho^{(t+1)} = \frac{n_\varrho(\boldsymbol{\theta}^{(t)}, \boldsymbol{\lambda}^{(t)})}{\sum_{\varrho \in \mathscr{P}_{L_\varrho}} n_\varrho(\boldsymbol{\theta}^{(t)}, \boldsymbol{\lambda}^{(t)})}, \ \forall\, \varrho \in \mathscr{P}_L \tag{A.5}
$$

## B  Calculation of Significance Score

In order to compute $\psi_w$ efficiently through dynamic programming, we have

$$
\psi_w = -\sum_{j=1}^{n} \log\left(1 - r_{wj}\right),
$$

where

$$
\begin{aligned}
r_{wj} &= \mathbb{P}\left( w \sim B_j \mid T_j, \mathcal{D}; \hat{\boldsymbol{\theta}}, \hat{\boldsymbol{\lambda}} \right) \tag{B.1} \\
&= \sum_{B_j \in \mathcal{B}_j} I\left( w \sim B_j \right) \cdot \mathbb{P}(B_j \mid T_j, \mathcal{D}; \hat{\boldsymbol{\theta}}, \hat{\boldsymbol{\lambda}}),
\end{aligned}
$$

with notation "$w \sim B_j$" meaning that word candidate $w$ appears in the segmented version of $T_j$ based on $B_j$.

## C Efficient Computation via Dynamic Programming

To implement the TopWORDS-Poetry algorithm, we need to calculate $n_{w,j}$ in (A.2), $n_{\varrho,j}$ in (A.3), $r_{w,j}$ in (B.1) and $\gamma_{jl}$ in (11) for $\forall\, T_j \in \mathcal{T}$.

For a specific $T_j = T = a_1 \cdots a_L$, by defining $T_{[t:s]} = a_t \cdots a_s$, we can shown that $n_{w,j}$, $n_{\varrho,j}$, $r_{wj}$ and $\gamma_{jl}$, which are all functions of $T_j$, have the following recursive formula:

$$n_w(T) = \frac{1}{p(T)} \sum_{\varrho \in \mathscr{P}_{L_j}} \lambda_\varrho \cdot n_w^\varrho(T),$$

$$n_w^\varrho(T) = \sum_{1 \le t < s \le L} \Big[ p^{(\varrho)}(T_{[<t]}) \cdot p^{(\varrho)}(T_{[>s]}) \\ \cdot \theta_w \cdot \prod_{t \le l < s} (1 - \rho_{\varrho l}) \cdot \rho_{\varrho s} \cdot I_{(T_{[t:s]} = w)} \Big],$$

$$n_\varrho(T) = \frac{\lambda_\varrho \cdot p^{(\varrho)}(T)}{p(T)},$$

$$r_w(T) = \frac{1}{p(T)} \sum_{\varrho \in \mathscr{P}_{L_j}} \lambda_\varrho \cdot r_w^\varrho(T),$$

$$r_w^\varrho(T) = \sum_{t=1}^{\tau_L} \Big[ r_w^\varrho(T_{[>t]}) \cdot I_{(T_{[1:t]} \ne w)} + I_{(T_{[1:t]} = w)} \Big] \\ \cdot \theta_{T_{[\le t]}} \cdot \prod_{1 \le l < t} (1 - \rho_{\varrho l}) \cdot \rho_{\varrho t} \cdot \frac{p^{(\varrho)}(T_{[>t]})}{p^{(\varrho)}(T)},$$

$$\gamma_l(T) = \frac{\sum_{\varrho \in \mathscr{P}_{L_j}} \lambda_\varrho \cdot p(T_{[\le l]}) \cdot p(T_{[>l]})}{p(T)},$$

where

$$p^{(\varrho)}(T_{[t:s]}) = \sum_{B \in \mathcal{B}_{[t:s]}} \mathbb{P}(T_{[t:s]} \mid \mathcal{D}; B, \boldsymbol{\theta}) \cdot \pi(B \mid \rho_{\varrho j}),$$

$$p(T) = \sum_{\varrho \in \mathcal{C}} \lambda_\varrho \cdot p^{(\varrho)}(T),$$

with $\mathcal{B}_{[t:s]}$ being the collection of all possible word boundary profiles of $T_{j_{[t:s]}}$.

We can compute $p^{(\varrho)}(T_{[<t]})$ and $p^{(\varrho)}(T_{[>t]})$ in linear time via dynamic programming to implement all computation issues in the following recursion:

$$p^{(\varrho)}(T_{[<t]}) = \sum_{1 \le s \le \min(t-1, \tau_L)} \Big[ p^{(\varrho)}(T_{[<t-s]}) \\ \cdot \theta_{T_{[t-s:t-1]}} \cdot \prod_{t-s \le l < t-1} (1 - \rho_{\varrho l}) \cdot \rho_{(\varrho), t-1} \Big],$$

$$p^{(\varrho)}(T_{[>t]}) = \sum_{1 \le s \le \min(L-t, \tau_L)} \Big[ p^{(\varrho)}(T_{[>t+s]}) \\ \cdot \theta_{T_{[t+1:t+s]}} \cdot \prod_{t+1 \le l < t+s} (1 - \rho_{\varrho l}) \cdot \rho_{\varrho, t+s} \Big],$$

## D Insights behind the Mathematical Formulation

The mathematical formulation of TopWORDS-Poetry is composed of four parts. The first part, which contains just formula (1), defines the basic Unigram Language Model (ULM) which serves as the backbone of the TopWORDS-series methods. Formula (2)-(6) form the second part, which defines the informative prior distribution for word boundary based on the potential segmentation patterns of poetry lines. The third part composed by formula (7)-(11) provides detailed results about Bayesian inference of the unknown model parameters and word boundary prediction under the ULM and the specified prior. And, the fourth part composed by formula (A.1)-(B.1) in the Appendix provides more computational details about the EM algorithm and significant score.

Among these four parts, part 1, 3 and 4 are all standard mathematical formulas about ULM, Bayesian inference and EM algorithm with fast calculation via dynamic programming, which are very similar to their counterparts in the TopWORDS-Seg algorithm proposed by Pan et al. (2022). The design of prior distribution in the second part, i.e., formula (2)-(6), however, makes a key difference and plays an essential role in this study. Among these formulas, the conjugate Dirichlet prior in formula (2) is a natural choice that has been adopted by Pan et al. (2022). Formula (3), however, specifies for a mixture prior distribution with multiple components to reflect the fact that each poetry line can potentially follow multiple segmentation patterns. Formula (4) corresponds to the fact that a segmentation pattern may not be exactly followed and we have a chance to break through the pattern template by placing a word boundary at each position randomly. Formula (5)- (6), on the other hand, specify the prior distribution for the relative

weights of the mixture components defined in formula (3). And, formula (7), at last, assembles all these piecewise prior distributions into a joint prior distribution to guide the learning.

## E More Details on TopWORDS-Poetry*

Let $(T_{2i-1}, T_{2i})$ be one couplet composed of two poetry lines. We can estimate the transition probability $\mathbb{P}(T_{2i} \mid T_{2i-1})$ based on couplets in training data $\mathcal{P}_{1000}$. Figure E1 visualizes the estimated transition probability matrices, suggesting there exists a positive correlation between the usage of segmentation patterns within a couplet.

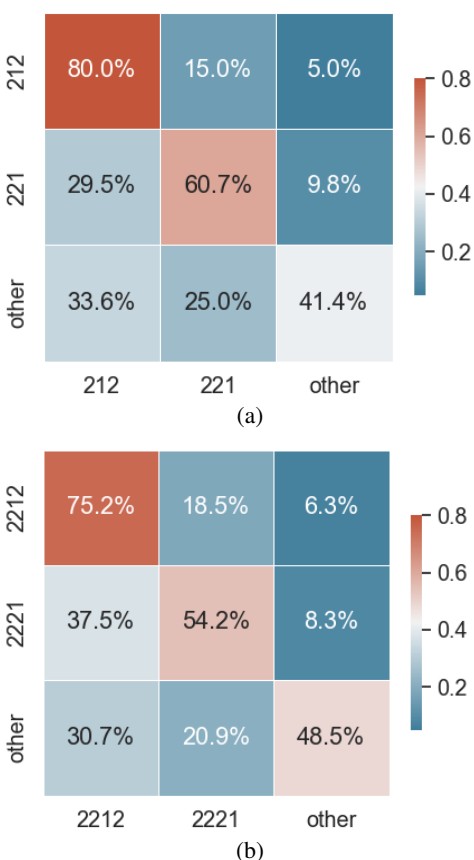

Figure E1: Transition probability matrix between patterns within one couplet on $\mathcal{T}_{1000}$. (a) Transition probability matrix for 5-character poem couplets. (b)Transition probability matrix for 7-character poem couplets.

These facts suggest that we can further improve TopWORDS-Poetry by modelling such correlation. In practice, this can achieved conveniently by merging two poetry lines in the same couplet into a longer concatenated line with 10 or 14 Chinese characters, and extending $\mathcal{P}_5$ and $\mathcal{P}_7$ to the following larger collections of longer segmentation patterns:

$$\mathcal{P}_{10} = \{\varrho_{212|212}, \varrho_{212|221}, \varrho_{221|212}, \varrho_{221|221}\},$$
$$\mathcal{P}_{14} = \{\varrho_{2212|2212}, \varrho_{2212|2221}, \varrho_{2221|2212},$$
$$\varrho_{2221|2221}\}. \qquad (E.1)$$

Segmentation patterns such as $\varrho_{212|212}$ means that the two poetry lines in one couplet have the same pattern 2-1-2, while $\varrho_{212|221}$ means that the two poetry lines have different patterns. Let $\boldsymbol{\lambda}_{10}$ and $\boldsymbol{\lambda}_{14}$ be the proportion vectors of $\mathcal{P}_{10}$ and $\mathcal{P}_{10}$, respectively. Replacing $\mathcal{P}_5$ and $\mathcal{P}_7$ by $\mathcal{P}_{10}$ and $\mathcal{P}_{10}$, we can implement TopWORDS-Poetry* by following exactly the same pipeline as in TopWORDS-Poetry, except for fixing $\rho_{\varrho, L_\varrho/2} = 1$ for any $\varrho \in \mathcal{P}_{10} \cup \mathcal{P}_{14}$ because there is always a word boundary between the two poetry lines in a couplet.

## F Details on Calling ChatGPT

To segment classical Chinese poetry lines by ChatGPT, we asked ChatGPT (model version: gpt-35-turbo) to segment all poetry lines in the testing dataset $\mathcal{P}_{1000}$ under the parameter setting where $temperature = 0.0$, $max\_tokens = 800$, $top\_p = 0.95$, $frequency\_penalty = 0$, $presence\_penalty = 0$ (all other parameters take the default values) with the prompt template in Figure F1.

Prompt template:

"对<>中的句子进行分词，用"/"隔开，不需要提供任何解释。
输入：<功蓋三分國。名成八陣圖。江流石不轉。遺恨失吞吳。>
回答：功蓋/三分國。名成/八陣圖。江流/石/不/轉。遺恨/失/吞吳。
输入：<[目标文本]>"

English translation:

"Please do text segmentation for the sentences in <>, separated by "/"", without providing any explanation.
Input: < His deeds covered a kingdom split in three, his fame completed the Plan of the Eight Formations. The River flows on, the stones do not turn, a remnant bitterness at his failure to swallow Wu.>
Answer: His deeds covered / a kingdom split in three /, his fame completed / the Plan of the Eight Formations. The River flows on /, the stones / do not / turn /, a remnant bitterness / at his failure / to swallow Wu.
Input:<[Target Text]>"

Figure F1: The prompt template for calling ChatGPT and its English translated version.

## G  Performance Evaluation on Word Discovery by Word Type

Performance evaluations on word discovery reported in Table 1 are pooled results for all word types including both common words and technical words, e.g., names, addresses and literary allusions. Here, we would like to report more detailed results for different types of technical words. For this purpose, we manually picked up all names, addresses and literary allusions in the 1000 segmented poems in the $\mathcal{P}_{1000}$, and summarized in Table G1 the performance of different approaches on discovering technical words of different types. From this table, we can see that TopWORDS-Poetry is effective in discovering all types of technical words, particularly for literary allusions that are often missed by competing methods.

| Method | Name | Adress | Allusion |
|---|---|---|---|
| Entity Count | 1303 | 924 | 723 |
| LTP | 20.64% | 22.29% | 19.78% |
| StanfordNLP | 50.42% | 52.71% | 44.67% |
| THULAC | 61.17% | 63.31% | 59.06% |
| PKUSEG | 63.70% | 64.94% | 60.86% |
| Jieba | 62.55% | 64.18% | 55.19% |
| TopWORDS | 84.96% | 84.20% | 82.99% |
| TopWORDS-Seg | 73.68% | 76.95% | 72.48% |
| RandomPattern | 80.05% | 76.84% | 69.85% |
| SelectPattern | 86.19% | 85.50% | 78.56% |
| ChatGPT | 80.97% | 83.23% | 78.98% |
| TopWORDS-Poetry | 87.18% | 88.64% | **88.38%** |
| TopWORDS-Poetry* | **87.72%** | **88.96%** | **88.38%** |

Table G1: Performance of different methods on discovering technical words of different types.

## H  Does TopWORDS-Poetry Perform Significantly Better?

To check whether TopWORDS-Poetry performs significantly better than competing methods, we conducted an additional experimental study based on the idea of bootstrapping (Efron and Tibshirani, 1994). For this purpose, we generated 20 pseudo testing datasets by randomly picking up 80% elements of $\mathcal{P}_{1000}$ without replacement. For each of these pseudo testing datasets, one version of the multi-dimensional performance measurements was calculated, resulting in an empirical distribution of performance measurements for every involved method. Table H1 summarizes the mean and standard deviation of the bootstrap distribution of each performance measurement for every involved method.

If TopWORDS-Poetry works significantly better than competing methods, we would like to observe an obvious shift in its empirical distributions with respect to the competing methods, which can be tested by a standard paired 2-sample $t$-test. The last two rows of Table H1 reports the performance margins between TopWORDS-Poetry* (the more robust member of the TopWORDS-Poetry family) and the best competing method outside the TopWORDS-Poetry family, and the $p$-values of the corresponding paired 2-sample $t$-test. Because most of these $p$-values are highly significant (<0.001), we are comfortable to claim that TopWORDS-Poetry indeed works better than existing methods in most performance evaluation dimensions. Although such a strategy is not very rigorous, it provides us with strong empirical evidence to gain useful insights about the statistical significance of TopWORDS-Poetry with respect to competing methods.

| | | Word discovery ($R_d^{(t,l)}$, $R_s^{(t,l)}$) | | | | | Text segmentation ($F_S$) | | | | |
|---|---|---|---|---|---|---|---|---|---|---|---|
| Method | $|V_d|$ | $R_d^{(2,2)}$ | $R_d^{(2,3)}$ | $|V_s|$ | $R_s^{(2,2)}$ | $R_s^{(2,3)}$ | $\mathcal{P}_{1000}$ | $\mathcal{P}_{250}^{4\times5}$ | $\mathcal{P}_{250}^{8\times5}$ | $\mathcal{P}_{250}^{4\times7}$ | $\mathcal{P}_{250}^{8\times5}$ |
| LTP | 39K | 60.39% ± 0.19% | 58.45% ± 1.47% | 2088 ± 26 | 19.08% ± 0.12% | 29.94% ± 1.05% | 20.05% ± 0.15% | 21.18% ± 0.80% | 18.78% ± 0.31% | 21.83% ± 0.33% | 19.67% ± 0.30% |
| StanfordNLP | 77K | 77.54% ± 0.23% | 63.71% ± 1.90% | 4921 ± 24 | 43.19% ± 0.25% | 44.61% ± 1.53% | 39.45% ± 0.18% | 40.33% ± 0.70% | 40.16% ± 0.41% | 41.18% ± 0.48% | 37.74% ± 0.32% |
| THULAC | 76K | 77.69% ± 0.21% | 43.62% ± 1.40% | 5502 ± 25 | 52.88% ± 0.22% | 34.22% ± 1.27% | 49.11% ± 0.15% | 50.08% ± 0.79% | 50.16% ± 0.31% | 49.43% ± 0.40% | 47.84% ± 0.23% |
| PKUSEG | 83K | 82.08% ± 0.15% | 48.36% ± 1.49% | 6318 ± 24 | 58.88% ± 0.23% | 36.16% ± 1.30% | 52.23% ± 0.17% | 52.48% ± 0.57% | 52.92% ± 0.41% | 52.92% ± 0.41% | 51.30% ± 0.36% |
| Jieba | 71K | 78.61% ± 0.15% | 43.80% ± 1.27% | 5677 ± 22 | 57.29% ± 0.21% | 37.78% ± 1.37% | 53.66% ± 0.14% | 56.50% ± 0.54% | 52.59% ± 0.36% | 54.31% ± 0.31% | 53.08% ± 0.29% |
| TopWORDS | 54K | 78.70% ± 0.25% | 90.15% ± 0.94% | 7080 ± 27 | 71.53% ± 0.29% | 89.84% ± 1.01% | 63.68% ± 0.21% | 63.31% ± 0.46% | 62.34% ± 0.32% | 64.99% ± 0.50% | 64.17% ± 0.32% |
| TopWORDS-Seg | 86K | 83.51% ± 0.14% | 54.05% ± 1.51% | 7106 ± 19 | 67.88% ± 0.25% | 45.20% ± 1.54% | 60.75% ± 0.20% | 60.01% ± 0.50% | 61.34% ± 0.31% | 60.53% ± 0.43% | 60.72% ± 0.40% |
| RandomPattern | 96K | 95.24% ± 0.11% | 0.00% ± 0.00% | 8015 ± 24 | 81.26% ± 0.23% | 0.00% ± 0.00% | 69.11% ± 0.22% | 63.78% ± 0.47% | 64.74% ± 0.40% | 72.72% ± 0.55% | 72.56% ± 0.23% |
| SelectPattern | 73K | 94.33% ± 0.13% | 0.00% ± 0.00% | 7688 ± 21 | 88.89% ± 0.14% | 0.00% ± 0.00% | 84.11% ± 0.17% | 81.14% ± 0.53% | 81.53% ± 0.27% | 86.48% ± 0.33% | 85.97% ± 0.26% |
| TopWORDS-Poetry | 82K | 96.32% ± 0.06% | 84.47% ± 1.08% | 7803 ± 21 | 89.09% ± 0.20% | 63.28% ± 1.64% | 84.50% ± 0.14% | 81.53% ± 0.46% | 81.91% ± 0.26% | 86.47% ± 0.42% | 86.54% ± 0.20% |
| TopWORDS-Poetry* | 82K | 96.39% ± 0.06% | 83.57% ± 1.24% | 7828 ± 24 | 89.47% ± 0.17% | 57.07% ± 1.57% | 84.73% ± 0.17% | 80.72% ± 0.45% | 82.67% ± 0.30% | 86.82% ± 0.42% | 86.69% ± 0.21% |
| Performance margin | - | 1.15 | -6.58 | - | 0.58 | -32.77 | 0.62 | -0.42 | 1.14 | 0.34 | 0.73 |
| $p$-value | - | $9.57 \times 10^{-23}$ | $2.22 \times 10^{-19}$ | - | $4.10 \times 10^{-13}$ | $1.15 \times 10^{-28}$ | $1.06 \times 10^{-12}$ | $2.25 \times 10^{-3}$ | $2.80 \times 10^{-17}$ | $7.85 \times 10^{-6}$ | $6.27 \times 10^{-12}$ |

Table H1: Performance of TopWORDS-Poetry for classical Chinese poetry by bootstrapping on word discovery and text segmentation compared to competing methods.