# OpenReview forum: "TopWORDS-Poetry: Simultaneous Text Segmentation and Word Discovery for Classical Chinese Poetry via Bayesian Inference"
_EMNLP/2023/Conference — EMNLP 2023 Main_

### Official Review · Reviewer_BAFH · 2023-08-04

**Soundness:** 4

**Excitement:**

3: Ambivalent: It has merits (e.g., it reports state-of-the-art results, the idea is nice), but there are key weaknesses (e.g., it describes incremental work), and it can significantly benefit from another round of revision. However, I won't object to accepting it if my co-reviewers champion it.

**Missing References:**

There is a lot of work on Bayesian word segmentation which the authors do not cite
https://www.semanticscholar.org/paper/A-Bayesian-framework-for-word-segmentation%3A-the-of-Goldwater-Griffiths/46f31f9069bb934498a288126053bcab01ff34aa
https://aclanthology.org/P09-1012/
https://link.springer.com/article/10.1007/s11168-011-9074-5

**Paper Topic And Main Contributions:**

This paper focuses on devising tools for processing classical Chinese poems. It is argued that standard Chinese processing tools are not directly applicable to classical poems as they contain many new words, named entities, and poetry specific patterns. The authors present a Bayesian unsupervised method of discovering new words in classical Chinese poems --- it is essentially a word segmentation problem. The authors propose an improvement of the method of Pan et al. (2022). The proposed method starts with a heuristic segmentation based on an initial word dictionary and the most frequent segmentation patterns. Prior distributions are then specified  based
given segmentation patterns;  a word discovery stage estimates word boundaries  and prunes the initial word dictionary D (see  Figure 2 for details). The target posterior is estimated by introducing latent variables and using an EM-style algorithm (derived in Appendix A).

**Questions For The Authors:**

- Please explain why the segmentation tool you are developing is required. In particular who is it for? Is it for scholars studying these
poems? Would the scholars have issues with unknown words and with segmentation? I would have thought it is their job to understand these poems. So if it is not scholars you are targetting who is it? Do you want to help lay people understand these poems? Is it common? Or you are developing a literary analysis tool?
- I am having trouble understanding how word discovery is different from text segmentation. Isn't segmentation in Chinese inserting boundaries between characters which would in turn give you word discovery?
- Can you please explain clarify part of your model is different from Peng et al (2022). Is it a prior that also takes into account segmentation patterns? In other words equation (3)?
- It would be good to report aggregate scores in Figure 3(a) and also whether differences between TopWORDS-Poetry(*) and the next best comparison system are statistically significant.
- did you try to see how a GPT style model performs your task using in context learning?

**Reasons To Accept:**

- The authors present a Bayesian unsupervised model of word segmentation which is a nice respite from supervised neural network models
- The model might be useful of other segmentation tasks (e.g., for documents)
- The paper servers as a good introduction to unsupervised word segmentation

**Reasons To Reject:**

- Slightly impenetrable, very young readers will struggle with the Bayesian formulation as formulas are presented without any intution
- The task and model are rather niche
- Added value on Peng et al. (2002) is unclear to me. The current model seems like an extension

**Reproducibility:**

3: Could reproduce the results with some difficulty. The settings of parameters are underspecified or subjectively determined; the training/evaluation data are not widely available.

**Reviewer Confidence:**

3: Pretty sure, but there's a chance I missed something. Although I have a good feel for this area in general, I did not carefully check the paper's details, e.g., the math, experimental design, or novelty.

---

> ### Author Rebuttal · Authors · 2023-08-29
>
> Many thanks for your insightful comments and detailed suggestions, which have stimulated us to think in depth about the academic positioning and representation of this paper. We fully agree with you that the proposed TopWORDS-Poetry method is an extension of the TopWORDS-Seg method in Pan et al. (2022), aiming at a niche task. But we believe that the research in this study still makes significant academic contributions to the NLP community, since classical Chinese poetry covers a body of texts with very unique features for which little research has been done. First, by formulating potential segmentation patterns of poetry lines as informative prior to guide text segmentation and word discovery under the framework of TopWORDS, we come up with an effective approach to process classical Chinese poetry and fill a blind spot in NLP literature. Second, experiments in this study confirm that existing methods established for general Chinese texts have various limitations in processing classical Chinese poetry. Third, applying TopWORDS-Poetry to the nearly 50,000 poems in the *Complete Tang Poetry*, we came up with a large collection of discovered words/phrases containing many named entities and literary allusions that have not been reported before, which may greatly promote the study of classical Chinese poetry via the paradigm of digital humanity. Moreover, the extension from TopWORDS-Seg to TopWORDS-Poetry is non-trivial, as the design of informative prior to count for the segmentation patterns of poetry lines involves deliciated modelling techniques on top of deep insights into the unique task.
>
> We totally agree that the Bayesian formulation involved in this paper is not friendly for readers who are not familiar with Bayesian inference. We are terribly sorry that we were not able to provide sufficient explanation about the intuition behind the mathematical formulas involved in this paper due to the limited space of the conference paper. In fact, the mathematical formulation of this paper is composed of four parts. The first part, which contains just formula (1), defines the basic *Unigram Language Model* (ULM) which serves as the backbone of the TopWORDS-series methods. Formula (2)-(6) form the second part, which defines the informative prior distribution for word boundary based on the potential segmentation patterns of poetry lines. The third part composed by formula (7)-(11) provides detailed results about Bayesian inference of the unknown model parameters and word boundary prediction under the ULM and the specified prior. And, the fourth part composed by formula (12)-(17) in the Appendix provides more computational details about the EM algorithm and significant score.
>
> Among these four parts, part 1, 3 and 4 are all standard mathematical formulas about ULM, Bayesian inference and EM algorithm with fast calculation via dynamic programming, which are very similar to their counterparts in the TopWORDS-Seg algorithm proposed by Pan et al. (2022). The design of prior distribution in the second part, i.e., formula (2)-(6), however, makes a key difference and plays an essential role in this study. Among these formulas, the conjugate Dirichlet prior in formula (2) is a natural choice that has been adopted by Pan et al. (2022). Formula (3), however, specifies for $B_j$ a mixture prior distribution with multiple components to reflect the fact that each poetry line can potentially follow multiple segmentation patterns. Formula (4) corresponds to the fact that a segmentation pattern may not be exactly followed and we have a chance to break through the pattern template by placing a word boundary at each position randomly. Formula (5)-(6), on the other hand, specify the prior distribution for the relative weights of the mixture components defined in formula (3). And, formula (7), at last, assembles all these piecewise prior distributions into a joint prior distribution to guide the learning. We would be very happy to explain the insights by uploading an additional supplementary material for this issue once the paper is fortunately accepted.
> Below are our point-to-point responses to your detailed questions.
>
>
>
> **Q1**. Please explain why the segmentation tool you are developing is required. In particular who is it for? Is it for scholars studying these poems? Would the scholars have issues with unknown words and with segmentation? I would have thought it is their job to understand these poems. So if it is not scholars you are targetting who is it? Do you want to help lay people understand these poems? Is it common? Or you are developing a literary analysis tool?
>
> **Response**：Thanks for raising this important question, which we did not discuss sufficiently in the current manuscript. The primary goal of this study is to develop a powerful tool for literary scholars to analyze classical Chinese poetry at a large scale via distance reading, a research paradigm that has been widely adopted by the community of digital humanity. Under the traditional paradigm of literary research, an individual scholar typically conducts an in-depth study on a small collection of poems written by specific poets, in particular periods, or in similar styles. Few efforts have been made to study a large number of poems at once from a global perspective due to the lack of technical tools that can effectively process classical Chinese poetry. In the history of Chinese literature, approximately 800,000 classical poems have been passed down, including 50,000 poems in *Complete Tang Poetry*. Although such a corpus is not as big as Wikipedia, it definitely exceeds the capability limit of individual scholars, leading to a great appeal for developing effective algorithms to process Chinese poetry texts. At present, there is still no widely adopted method for such a purpose. Once we make a technological breakthrough in this direction, we would expect an exciting interaction between research on classical Chinese poetry and NLP technology, which may lead to a new era of literary study and promote many exciting applications for cultural education and dissemination, helping more people to better enjoy the charm of classical poetry. The proposed method has the potential to play such a role.
>
> **Q2**. I am having trouble understanding how word discovery is different from text segmentation. Isn't segmentation in Chinese inserting boundaries between characters which would in turn give you word discovery?
>
> **Response**: Sorry for not discussing this important issue sufficiently in the current manuscript, as we believe that Pan et al. (2022) have already provided an insightful discussion on this important issue. At the conceptual level, word discovery and text segmentation are two highly related tasks in Chinese NLP, because inserting word boundaries between Chinese characters can naturally generate the segmented words. In practice, however, as pointed out by Pan et al. (2022), these two tasks have different focuses and often entangle with each other. This is because classical methods for Chinese text segmentation usually require the support of a preloaded vocabulary, which is either an explicit list of words, or an implicit word collection encoded in the large language models. When the preloaded vocabulary fits the target corpus well, we typically expect reasonably good results on text segmentation. Once the preloaded vocabulary fails to cover the underlying vocabulary of the target corpus, however, we often observe significant performance degeneration on text segmentation, as demonstrated in Pan et al. (2022). The task in this study happens to fall into the second case, as classical Chinese poetry contains many unique named entities and literary allusions that never appear in general texts. In this scenario, we would have to concern the word discovery task and consider it with the text segmentation task together in pursuit of high-quality results.
>
> **Q3**. Can you please explain clarify part of your model is different from Peng et al (2022). Is it a prior that also takes into account segmentation patterns? In other words equation (3)?
>
> **Response**: We believe that the paper you mentioned should be Pan et al. (2022). Indeed, as an extension of the key ideas in Pan et al. (2022), TopWORDS-Poetry is very similar to TopWORDS-Seg in model structure and learning strategy. The key difference is that TopWORDS-Poetry introduces a more complicated mixture prior distribution to count for the potential segmentation patterns for Chinese poetry lines, which also leads to new computational challenges on parameter estimation, model selection and post Bayesian inference.
>
>
> **Q4**. It would be good to report aggregate scores in Figure 3(a) and also whether differences between TopWORDS-Poetry(*) and the next best comparison system are statistically significant.
>
> **Response**: Thanks for these constructive suggestions. We totally agree that it would be ideal if we can integrate the multi-dimensional evaluation measurements into a one-dimensional score. To the best of our efforts, however, we found that it’s non-trivial to define such an integrated score due to the absence of a reasonable criterion to conduct the integration. Totally, two types of evaluation measurements are reported in Fig 3 (a): one for the task of word discovery, and one for the task of text segmentation. Because the two tasks are quite different in nature, there does not exist a straightforward way to combine their performance measurements. Although trivial aggregation such as simple or weighted averaging is always possible, it would be very difficult to interpret the averaged value obtained in this way and determine the weights wisely. Thus, to avoid potential confusions and disputes, we chose to report detailed, instead of integrated, results.
>
> To evaluate the statistically significance you suggested, we conducted an additional experimental study based on the idea of bootstrapping. To be concrete, generated 20 pseudo testing datasets based on the 1000 manually segmented poems in original testing dataset $P_{1000}$ by randomly picking up 80% elements of $P_{1000}$. For each of these pseudo testing datasets, we can calculate one version of the multi-dimensional performance measurements, resulting in an empirical distribution of performance measurements for every involved method. If TopWORDS-Poetry works significantly better than competing methods, we would like to observe an obvious shift in its empirical distributions with respect to the competing methods, which can be tested by a standard paired 2-sample t-test. After implementing such a protocol, we found that the p-value of the paired 2-sample t-test is highly significant (<0.001) in almost all dimensions, indicating that TopWORDS-Poetry indeed works better than existing methods. Although such a strategy is not very rigorous from the theoretical point of view, it provides us with empirical evidence to gain insights about the statistical significance of the results by TopWORDS-Poetry with respect to the competing methods. We hope these efforts can help ease your concerns.
> Detailed performance (mean and standard error) with differences from the next best methods and p-values by paired 2-sample t-test are shown below:
>
> | Method | $R_d^{(t,l)}$ |  |  | $R_s^{(t,l)}$ |  |  | $F_S$ |  |  |  |   |
> |---|---|---|---|---|---|---|---|---|---|---|---|
> |  | $\|V_d\|$ | $R_d^{(2,2)}$ | $R_d^{(2,3)}$ | $\|V_s\|$ | $R_s^{(2,2)}$ | $R_s^{(2,3)}$ | $P_{1000}$ | $P_{250}^{4\times5}$ | $P_{250}^{8\times5}$ | $P_{250}^{4\times7}$ | $P_{250}^{8\times7}$  |
> | LTP | 39K | 60.39% ± 0.19% | 58.45% ± 1.47% | 2088 ± 26 | 19.08% ± 0.12% | 29.94% ± 1.05% | 20.05% ± 0.15% | 21.18% ± 0.8% | 18.78% ± 0.31% | 21.83% ± 0.33% | 19.67% ± 0.3%  |
> | StanfordNLP | 77K | 77.54% ± 0.23% | 63.71% ± 1.9% | 4921 ± 24 | 43.19% ± 0.25% | 44.61% ± 1.53% | 39.45% ± 0.18% | 40.33% ± 0.7% | 40.16% ± 0.41% | 41.18% ± 0.48% | 37.74% ± 0.32%  |
> | THULAC | 76K | 77.69% ± 0.21% | 43.62% ± 1.4% | 5502 ± 25 | 52.88% ± 0.22% | 34.22% ± 1.27% | 49.11% ± 0.15% | 50.08% ± 0.79% | 50.16% ± 0.31% | 49.43% ± 0.4% | 47.84% ± 0.23%  |
> | PKUSEG | 83K | 82.08% ± 0.15% | 48.36% ± 1.49% | 6318 ± 24 | 58.88% ± 0.23% | 36.16% ± 1.3% | 52.23% ± 0.17% | 52.48% ± 0.57% | 52.92% ± 0.41% | 52.92% ± 0.41% | 51.3% ± 0.36%  |
> | Jieba | 71K | 78.61% ± 0.15% | 43.8% ± 1.27% | 5677 ± 22 | 57.29% ± 0.21% | 37.78% ± 1.37% | 53.66% ± 0.14% | 56.5% ± 0.54% | 52.59% ± 0.36% | 54.31% ± 0.31% | 53.08% ± 0.29%  |
> | TopWORDS | 54K | 78.7% ± 0.25% | **90.15%** ± 0.94% | 7080 ± 27 | 71.53% ± 0.29% | **89.84%** ± 1.01% | 63.68% ± 0.21% | 63.31% ± 0.46% | 62.34% ± 0.32% | 64.99% ± 0.5% | 64.17% ± 0.32%  |
> | TopWORDS-Seg | 86K | 83.51% ± 0.14% | 54.05% ± 1.51% | 7106 ± 19 | 67.88% ± 0.25% | 45.2% ± 1.54% | 60.75% ± 0.2% | 60.01% ± 0.5% | 61.34% ± 0.31% | 60.53% ± 0.43% | 60.72% ± 0.4%  |
> | RandomPattern | 96K | 95.24% ± 0.11% | 0.0% ± 0.0% | 8015 ± 24 | 81.26% ± 0.23% | 0.0% ± 0.0% | 69.11% ± 0.22% | 63.78% ± 0.47% | 64.74% ± 0.4% | 72.72% ± 0.55% | 72.56% ± 0.23%  |
> | SelectPattern | 73K | 94.33% ± 0.13% | 0.0% ± 0.0% | 7688 ± 21 | 88.89% ± 0.14% | 0.0% ± 0.0% | 84.11% ± 0.17% | 81.14% ± 0.53% | 81.53% ± 0.27% | 86.48% ± 0.33% | 85.97% ± 0.26%  |
> | TopWORDS-Poetry | 82K | 96.32% ± 0.06% | 84.47% ± 1.08% | 7803 ± 21 | 89.09% ± 0.2% | 63.28% ± 1.64% | 84.5% ± 0.14% | **81.53%** ± 0.46% | 81.91% ± 0.26% | 86.47% ± 0.42% | 86.54% ± 0.2%  |
> | TopWORDS-Poetry $^*$ | 82K | **96.39%** ± 0.06% | 83.57% ± 1.24% | 7828 ± 24 | **89.47%** ± 0.17% | 57.07% ± 1.57% | **84.73%** ± 0.17% | 80.72% ± 0.45% | **82.67%** ± 0.3% | **86.82%** ± 0.42% | **86.69%** ± 0.21%  |
> | Difference between TopWORDS-Poetry$^*$ and next best method |  | 1.15 | -6.58 |  | 0.58 | -32.77 | 0.62 | -0.42 | 1.14 | 0.34 | 0.73  |
> | p value by paired 2-sample t-test  |  | 9.57E-23 | 2.22E-19 |  | 4.10E-13 | 1.15E-28 | 1.06E-12 | 2.25E-03 | 2.80E-17 | 7.85E-06 | 6.27E-12  |
>
>
> **Q5**. did you try to see how a GPT style model performs your task using in context learning?
>
> **Response**：Thanks for this insightful suggestion! To evaluate the performance of GPT-style models on segmenting classical Chinese poetry, we asked ChatGPT (model version: gpt-35-turbo) to segment all poetry lines in the testing dataset $P_{1000}$ under the following prompt template:
>
> "请对<>中的句子进行分词，用"/"隔开，不需要提供任何解释。
> 输入：<功蓋三分國。名成八陣圖。江流石不轉。遺恨失吞吳。>
> 回答：功蓋/三分國。名成/八陣圖。江流/石/不/轉。遺恨/失/吞吳。
> 输入：<[目标文本]>"
>
> Below is the English translation of the above prompt template:
>
> “Please do text segmentation for the sentences in <>, separated by "/", without providing any explanation.
> Input: < His deeds covered a kingdom split in three, his fame completed the Plan of the Eight Formations. The River flows on, the stones do not turn, a remnant bitterness at his failure to swallow Wu.>
> Answer: His deeds covered / a kingdom split in three /, his fame completed / the Plan of the Eight Formations. The River flows on /, the stones / do not / turn /, a remnant bitterness / at his failure / to swallow Wu.
> Input:<[Target Text]>
>
> | Method | $R_s^{(t,l)}$ |  |  | $F_S$ |  |  |  |   |
> |---|---|---|---|---|---|---|---|---|
> |  | $\|V_s\|$ | $R_s^{(2,2)}$ | $R_s^{(2,3)}$ | $P_{1000}$ | $P_{250}^{4\times5}$ | $P_{250}^{8\times5}$ | $P_{250}^{4\times7}$ | $P_{250}^{8\times7}$  |
> | LTP | 2.5K | 19.28% | 29.94% | 20.00% | 21.16% | 18.72% | 21.72% | 19.65%  |
> | StanfordNLP | 6.0K | 43.56% | 45.54% | 39.49% | 40.45% | 40.26% | 41.07% | 37.80%  |
> | THULAC | 6.7K | 53.28% | 35.03% | 49.08% | 50.27% | 50.14% | 49.25% | 47.78%  |
> | PKUSEG | 7.6K | 59.26% | 36.31% | 52.17% | 52.62% | 52.96% | 52.65% | 51.20%  |
> | Jieba | 6.8K | 57.30% | 37.58% | 53.62% | 56.40% | 52.60% | 54.23% | 53.04%  |
> | TopWORDS | 8.5K | 71.22% | **89.49%** | 63.71% | 63.34% | 62.49% | 65.08% | 64.07%  |
> | TopWORDS-Seg | 8.6K | 68.06% | 44.90% | 60.67% | 60.26% | 61.27% | 60.35% | 60.55%  |
> | RandomPattern | 9.7K | 81.61% | 0.00% | 69.09% | 63.62% | 64.70% | 72.80% | 72.56%  |
> | SelectPattern | 9.2K | 88.71% | 0.00% | 84.12% | 81.27% | 81.57% | 86.46% | 85.91%  |
> | TopWORDS-Poetry | 9.3K | 88.98% | 62.74% | 84.48% | **81.50%** | 81.95% | 86.49% | 86.46%  |
> | TopWORDS-Poetry $^*$ | 9.4K | **89.36%** | 56.69% | **84.71%** | 80.68% | **82.70%** | **86.85%** | **86.63%**  |
> | **ChatGPT** | 8.0K | 77.12% | 66.24% | 73.48% | 72.19% | 73.17% | 71.99% | 74.96%  |
>
>
> After summarizing the results output by ChatGPT, we found that the $R_S$ and $F_S$ scores of ChatGPT concentrate in a narrow region of around 70%, leaving a wide performance margin (>10%) with respect to TopWORDS-Poetry. Such results confirm that TopWORDS-Poetry achieves the SOTA on processing classical Chinese poetry, even when GPT-style models are taken into consideration.
>
>
> **Q6**. There is a lot of work on Bayesian word segmentation which the authors do not cite https://www.semanticscholar.org/paper/A-Bayesian-framework-for-word-segmentation%3A-the-of-Goldwater-Griffiths/46f31f9069bb934498a288126053bcab01ff34aa https://aclanthology.org/P09-1012/ https://link.springer.com/article/10.1007/s11168-011-9074-5
>
> **Response**：Thanks for listing these references missing in the current manuscript. Based on our investigation, these references have been well discussed by Pan et al. (2022). We would add these references into the manuscript with discussions if the paper is fortunately accepted.

---

### Official Review · Reviewer_qve5 · 2023-08-07

**Soundness:** 4

**Excitement:**

3: Ambivalent: It has merits (e.g., it reports state-of-the-art results, the idea is nice), but there are key weaknesses (e.g., it describes incremental work), and it can significantly benefit from another round of revision. However, I won't object to accepting it if my co-reviewers champion it.

**Paper Topic And Main Contributions:**

This paper presents an unsupervised Bayesian model to jointly perform text segmentation and word discovery (i.e. automatic vocabulary induction). The method is based on an existing Bayesian method (Pan et al., 2022) with some adaptations specifically tailored towards the specific genre of classical Chinese poetry. One of the key ideas is to exploit the regularity of the poetry's meter, modeled as a Bayesian prior. The method yields good results in both segmentation and vocabulary induction.

**Reasons To Accept:**

* a novel method for joint text segmentation and vocabulary induction in classic Chinese poetry; even though the method is to a large extent based on an existing model, there is sufficient novelty in the adaptations to the specific genre, which I find interesting

* authors provide an extensive evaluation and comparison to related work

**Reasons To Reject:**

* the application of the method is fairly limited

* descriptions are fairly technical; technical adequacy is a good thing, but I feel some more effort could have been taken to better explain the reasoning behind the formulae


**Reproducibility:**

3: Could reproduce the results with some difficulty. The settings of parameters are underspecified or subjectively determined; the training/evaluation data are not widely available.

**Reviewer Confidence:**

4: Quite sure. I tried to check the important points carefully. It's unlikely, though conceivable, that I missed something that should affect my ratings.

---

> ### Author Rebuttal · Authors · 2023-08-29
>
> Many thanks for your sharp comments and constructive suggestions. Below are our point-to-point responses to your comments.
>
> We fully agree with you that the proposed TopWORDS-Poetry is indeed a unique approach designed for a very specific application. But we believe that the research in this study still makes significant academic contributions to the NLP community, since classical Chinese poetry covers a body of texts with very unique features for which little research has been done. First, by formulating potential segmentation patterns of poetry lines as informative prior to guide text segmentation and word discovery under the framework of TopWORDS, we come up with an effective approach to process classical Chinese poetry and fill a blind spot in NLP literature. Second, experiments in this study confirm that existing methods established for general Chinese texts have various limitations in processing classical Chinese poetry. Third, applying TopWORDS-Poetry to the nearly 50,000 poems in the *Complete Tang Poetry*, we came up with a large collection of discovered words/phrases containing many named entities and literary allusions that have not been reported before, which may greatly promote the study of classical Chinese poetry via the paradigm of digital humanity.
>
> We are terribly sorry that we were not able to provide sufficient explanation about the intuition behind the mathematical formulas involved in this paper due to the limited space of the conference paper. We would be very happy to explain the reasoning behind the TopWORDS-Poetry algorithm by providing an additional supplementary material once the paper is accepted. In fact, the 17 major formulas in our paper can be divided into four groups. The first group, which contains just formula (1), defines the basic *Unigram Language Model* (ULM) which serves as the backbone of the TopWORDS-series methods. Formula (2)-(6) form the second group of formulas, which defines the informative prior distribution for word boundary based on the potential segmentation patterns of poetry lines. The third group composed of formula (7)-(11) provides detailed results about Bayesian inference of the unknown model parameters and word boundary prediction under the ULM and the specified prior. The fourth group composed by formula (12)-(17) in the Appendix provides more computational details about the EM algorithm and significant score.
>
> Among these four groups of formulas, group 1, 3 and 4 are all standard mathematical formulas about ULM, Bayesian inference and EM algorithm with fast calculation via dynamic programming, which are very similar to their counterparts in the TopWORDS-Seg algorithm proposed by Pan et al. (2022). Formulas in the second group, however, make a key difference and play an essential role in this study. Below, we explain in detail why we specify prior distributions as in formula (2)-(6). In fact, the conjugate Dirichlet prior in formula (2) is a natural choice that has been adopted by Pan et al. (2022). Formula (3), however, specifies for $B_j$ a mixture prior distribution with multiple components to reflect the fact that each poetry line can potentially follow multiple segmentation patterns. Formula (4) corresponds to the fact that a segmentation pattern may not be exactly followed and we have a chance to break through the pattern template by placing a word boundary at each position randomly. Formula (5)-(6), on the other hand, specify the prior distribution for the relative weights of the mixture components defined in formula (3). And, formula (7), at last, assembles all these piecewise prior distributions into a joint prior distribution to guide the learning.
>
> We hope the above explanations would help better clarify potential confusions. We would be more than happy to upload an additional supplementary material for this issue if our paper is fortunately accepted.

---

### Official Review · Reviewer_m3qs · 2023-08-13

**Soundness:** 4

**Excitement:**

4: Strong: This paper deepens the understanding of some phenomenon or lowers the barriers to an existing research direction.

**Paper Topic And Main Contributions:**

The paper describes TopWORDS-Poetry, a Bayesian text segmentation algorithm optimized for classical Chinese poetry. The method is an adaptation of TopWORDS-Seg (Pan et al. 2022) that incorporates genre-specific information about segmentation patterns into the prior distribution. Performance is evaluated on word discovery and text segmentation tasks using the Complete Tang Poetry corpus.


**Questions For The Authors:**

A: What might account for the large difference in R_s(2,3) between TopWORDS and TopWORDS-Poetry?

B: In their discussion of off-the-shelf segmentation tools, the authors write that "it is not realistic to tune these tools for this specific task, because no training corpus for classical Chinese poetry are publicly available yet, to the best of our knowledge." I was confused by this claim - why does Complete Tang Poetry not count as a suitable corpus in this context? I think it would be helpful to provide a short rationale for not attempting fine-tuning of a general-purpose tool and instead proceeding directly to creation of a bespoke method.

**Reasons To Accept:**

-The authors report a detailed empirical evaluation of TopWORDS-Poetry and show that it outperforms a range of other methods, including off-the-shelf word segmenters and TopWORDS-Seg.

-The method is well-described, and ample details are provided for replicating the analyses.

-Classical Chinese remains an under-resourced language, and the method should be immediately useful both for NLP researchers and for digital humanities scholars interested in the corpus. I appreciate that the authors have taken genre differences seriously and exploited the unique structure of classical Chinese verse to develop a bespoke segmentation tool.

**Reasons To Reject:**

I think the paper makes a strong contribution overall. My main suggestion for improvement would be to provide some finer-grained evaluation data. In particular, the authors emphasize that their method performs better with lexical items rarely found outside of poetic corpora, such as named entities and literary allusions. While the word clouds in Figure 3 are interesting, I would like to see more quantitative evidence for this claim, including performance information broken down by word or phrase type. How, for instance, does the tool do specifically on allusive passages? I appreciate that doing so would require additional manual work, but I think having a breakdown across a subset of the test set would be very informative.

Also, I would be interested to read further discussion about how the method might enable new studies of literary allusions. As the authors are no doubt aware, quantitative study of intertextuality is an topic of ongoing, cross-lingual interest in cultural analytics and digital humanities (e.g., Sturgeon 2018 Digital Scholarship in the Humanities, Burns et al. 2021 NAACL, Moritz et al. 2016 EMNLP), so such discussion might help excite and orient readers who are primarily interested in applications.

**Reproducibility:**

4: Could mostly reproduce the results, but there may be some variation because of sample variance or minor variations in their interpretation of the protocol or method.

**Reviewer Confidence:**

4: Quite sure. I tried to check the important points carefully. It's unlikely, though conceivable, that I missed something that should affect my ratings.

**Typos Grammar Style And Presentation Improvements:**

I would recommend splitting Figure 3 into multiple parts. (a) and (b) should be standalone tables, and the font size should be increased to improve readability. The word clouds in (c) - (e) can then be kept as Figure 3.

"Corpus" is singular, "corpora" plural, so, e.g., "no training corpus...are publicly available" is incorrect (lines 130-131).

Would it be possible to provide translations of the word clouds to improve accessibility of the paper for the full audience of EMNLP?

---

> ### Author Rebuttal · Authors · 2023-08-29
>
> Many thanks for your positive comments, constructive suggestions and insightful questions. Below are our point-to-point responses to your comments and questions.
>
> First, to provide more quantitative evidence to compare the performance of different approaches on discovering unique lexical terms such as named entities and literary allusions by word or phrase type as you suggested, we manually labelled all names, addresses and literary allusions in the 1000 manually segmented poems in the $P_{1000}$ data set, and summarized in the table below the performance of different approaches on discovering these lexical terms:
>
> | Method | Name | Adress | Allusion  |
> |---|---|---|---|
> | entity count | 1303  | 924  | 723   |
> | LTP | 20.64% | 22.29% | 19.78%  |
> | StanfordNLP | 50.42% | 52.71% | 44.67%  |
> | THULAC | 61.17% | 63.31% | 59.06%  |
> | PKUSEG | 63.70% | 64.94% | 60.86%  |
> | Jieba | 62.55% | 64.18% | 55.19%  |
> | TopWORDS | 84.96% | 84.20% | 82.99%  |
> | TopWORDS-Seg | 73.68% | 76.95% | 72.48%  |
> | RandomPattern | 80.05% | 76.84% | 69.85%  |
> | SelectPattern | 86.19% | 85.50% | 78.56%  |
> | TopWORDS-Poetry | 87.18% | 88.64% | **88.38%**  |
> | TopWORDS-Poetry^* | **87.72%** | **88.96%** | **88.38%**  |
>
> From this table, we can see that TopWORDS-Poetry is effective in discovering all types of lexical terms, particularly for literary allusions that are often missed by most competing methods. We would be very happy to add these new results to the manuscript as main texts or supplementary materials if our paper can be accepted by the conference.
>
> Second, in response to your request for more discussions about how the proposed method might enable new studies of literary allusions, we have carefully studied the references you recommended and come up with the insight that our work and these references have quite different focuses although they are in general related. Due to the high conciseness of classical Chinese poetry, in most cases, literature illusions in classical Chinese poetry are short phrases with no more than 4 Chinese characters highlighting particular events or literary imageries. For example, the literature illusion "老萊衣" (Laolai’s Clothes) refers to the colorful clothes worn by 老萊 (Laolai Zi), a hermit in the country of Chu in the Spring and Autumn period who was famous by amusing his parents wholeheartedly by wearing colorful clothes and making funny movements when he was 70 years old. In Chinese culture, "老萊衣" has become a symbol of filial piety, because the story behind the phrase indicates the spirit of sacrificing oneself for the benefit of parents, which is the key element of filial piety. Literature illusions like this one can deliver complicated meanings with concise texts, increasing the charm of classical Chinese poetry greatly. Intertextuality studied by Moritz et al. (2016), Sturgeon (2018) and Burns et al. (2021), however, mainly focuses on identifying parallel sentences or passages that are literally different but have similar meanings. According to Sturgeon (2018), below is a typical example of such parallel sentences:
>
> "使乌获疾引牛尾，尾绝力勯，而牛不可行，逆也" (If let the strongman Wu Huo pull the tail of the ox, and if the tail breaks and Wu Huo is exhausted but the ox does not move forward, it would be considered a reverse situation) in《吕氏春秋》(*Lvshi Chunqiu*);
>
> "今使乌获、藉蕃从后牵牛尾，尾绝而不从者，逆也" (Today, if let the strongman Wu Huo and the helper Jie Fan pull the tail of the ox from behind, and if the tail breaks and the ox does not move forward, it would be considered a reverse situation) in《淮南子》(*Huainanzi*).
>
> Apparently, the task of discovering parallel sentences/paragraphs is more like a matching problem based on sentence/paragraph meaning, which is different from the word/phrase discovery task in this paper. On the other hand, however, these two tasks are also closely related, if two parallel sentences/paragraphs correspond to the same literature illusion, which can be effectively summarized into a common short phrase representing the literature illusion. We would be very happy to cite the references you recommended in our paper with in-depth discussions, if our paper can be accepted by the conference.
>
> Third, in response to your question on the large difference in $R_s^{(2,3)}$ between TopWORDS and TopWORDS-Poetry, we would like to provide the following explanation. In general, TopWORDS is a more aggressive word discoverer and text segmenter than TopWORDS-Poetry, which tends to report more longer words/phrases in word discovery and reserve more longer words/phrases in text segmentation. Although such a property makes TopWORDS a powerful tool for Chinese word/phrase discovery in open domains, it often under-segments Chinese texts as pointed out by Pan et al. (2022). Similar to TopWORDS-Seg, the proposed TopWORDS-Poetry tries to re-balance the ability of TopWORDS on word discovery and text segmentation tasks by introducing informative prior information on text segmentation under the Bayesian framework. Such an effort would naturally enhance the performance of TopWORDS-Poetry on poetry line segmentation at the price of performance degeneration on word discovery. That is why we observe the large difference in $R_s^{(2,3)}$ between TopWORDS and TopWORDS-Poetry. In fact, a similar phenomenon has been reported by Pan et al. (2022). In practice, we can adjust the granularity of poetry line segmentation in TopWORDS-Poetry by selecting the hyper-parameters $\kappa_s$: if we increase $\kappa_s$ from the current 0.001 to 0.1, the $R_s^{(2,3)}$ of TopWORDS-Poetry would increase to 83%, a value very close to TopWORDS. But, if we do so, the $F_S$ scores of TopWORDS-Poetry would decrease from the current ~85% to ~80%, which is slightly smaller than the 84.12% of SelectPattern. In the manuscript, we choose to specify $\kappa_s=0.001$ instead of 0.1 to make sure that TopWORDS-Poetry can dominate SelectPattern in all dimensions. If we could accept a little bit of relaxation in the $F_S$ scores, we can easily get a $R_s^{(2,3)}$ that is as large as 80% by choosing $\kappa_s=0.1$ instead. We hope these explanations can help better clarify things.
>
> Fourth, about your question on why the *Complete Tang Poetry* itself cannot serve as a suitable training corpus to tune existing methods, we would like to respond as follows. To serve as a qualified training corpus to tune existing methods, a corpus must contain well-segmented Chinese texts with all word boundaries explicitly labelled. The original *Complete Tang Poetry*, however, only contains raw poetry lines without any word boundaries. That is why we cannot tune existing methods with the *Complete Tang Poetry* itself.
>
> At last, we would correct the typos you pointed out, split Fig 3 into separated parts as you suggested, and enhance the word clouds with English translations, once we get the chance to update the manuscript.

---

### Meta-Review · Area_Chair_4iLT · 2023-09-10

**Recommendation:** 5

**Metareview:**

This paper presents an unsupervised Bayesian model designed for text segmentation and word discovery. They extend an existing method, with the idea that the regularity of the poetry's meter can be incorporated as a Bayesian prior. The proposed method shows good performance both on word discovery and text segmentation.

The reviewers concur that the authors have introduced a clever idea that is empirically justified. Personally, I love to see a new Bayesian model in 2023. While younger readers may face challenges in comprehending the approach, this should not pose a barrier to its acceptance.

---

### Decision · Program_Chairs · 2023-10-07

**Decision:**

Accept-Main

**Comment:**

This paper presents an unsupervised Bayesian model designed for text segmentation and word discovery. They extend an existing method, with the idea that the regularity of the poetry's meter can be incorporated as a Bayesian prior. The proposed method shows good performance both on word discovery and text segmentation.

The reviewers concur that the authors have introduced a clever idea that is empirically justified. Personally, I love to see a new Bayesian model in 2023. While younger readers may face challenges in comprehending the approach, this should not pose a barrier to its acceptance.